# Structural basis of lipopolysaccharide assembly by the outer membrane translocon holo-complex

Haoxiang Chen[1,7], Axel Siroy[2,7], Violette Morales [1,7], Dominik Gurvič[3], Yves Quentin [1], Stephanie Balor[4], Yassin A. Abuta'a[1], Maurine Marteau[1,5,6], Carine Froment[5,6], Anne Caumont-Sarcos[1], Julien Marcoux [5,6], Phillip J. Stansfeld [3] ✉, Rémi Fronzes [2] ✉ & Raffaele Ieva [1] ✉

Lipopolysaccharide (LPS) assembly at the surfaces-exposed leaflet of the bacterial outer membrane (OM) is mediated by the OM LPS translocon. An essential transmembrane β-barrel protein, LptD, and a cognate lipoprotein, LptE, translocate LPS selectively into the OM external leaflet via a poorly understood mechanism. Here, we characterize two additional translocon subunits, the lipoproteins LptM and LptY (formerly YedD). We use single-particle cryo-EM analysis, functional assays and molecular dynamics simulations to visualize the roles of LptM and LptY at the translocon holo-complex LptDEMY, uncovering their impact on LptD conformational dynamics. Whereas LptY binds and stabilizes the periplasmic LptD β-taco domain that functions as LPS receptor, LptM intercalates the lateral gate of the β-barrel domain, promoting its opening and access by LPS. Remarkably, we demonstrate a conformational switch of the LptD β-taco/β-barrel interface alternating between contracted and extended states. β-strand 1 of LptD, which defines the mobile side of the lateral gate, binds LPS and performs a stroke movement toward the external leaflet during the contracted-to-extended state transition. Our findings support a detailed mechanistic framework explaining the selective transport of LPS to the membrane external leaflet.

Diderm bacteria are encased by an outer membrane (OM) that serves as a protective barrier and a platform for environmental interactions[1,2]. The OM is typically characterized by the presence of integral outer membrane proteins (OMPs), which fold into transmembrane β-barrels[3]. Some OMPs play crucial roles in OM biogenesis and molecular transport. Among these, the OMP foldase BamA (also known as subunit A of the β-barrel assembly machinery, BAM) and the transporter LptD are hallmark OMPs found in the vast majority of diderm bacteria, typically Gram-negative[4–6]. In Proteobacteria, LptD, together with its cognate lipoprotein LptE, forms the OM translocon responsible for inserting lipopolysaccharide (LPS) into the external leaflet[7–10]. This process establishes an asymmetric bilayer, with glycerophospholipids confined to the internal leaflet[11–13]. The asymmetric distribution of LPS is crucial for organizational and structural roles of

[1]Laboratoire de Microbiologie et Génétique Moléculaires (LMGM), Centre de Biologie Intégrative (CBI), Université de Toulouse, CNRS, Toulouse, France. [2]Microbiologie Fondamentale et Pathogénicité (MFP), Institut Européen de Chimie et Biologie, Université de Bordeaux, CNRS, Pessac, France. [3]School of Life Sciences and Department of Chemistry, Gibbet Hill Campus, University of Warwick, Coventry, UK. [4]METi, Centre de Biologie Intégrative (CBI), Université de Toulouse, CNRS, Toulouse, France. [5]Institut de Pharmacologie et de Biologie Structurale (IPBS), Université de Toulouse, CNRS, Toulouse, France. [6]Infrastructure Nationale de Protéomique, ProFI, UAR 2048, Toulouse, France. [7]These authors contributed equally: Haoxiang Chen, Axel Siroy, Violette Morales. ✉e-mail: phillip.stansfeld@warwick.ac.uk; remi.fronzes@u-bordeaux.fr; raffaele.ieva@utoulouse.fr

the OM, making it essential for the survival of most Gram-negative bacteria. Neighboring LPS molecules interact electrostatically via phosphate groups that are bridged by divalent cations. This tightly packed molecular arrangement forms a membrane structure that is not only impermeable to many antibiotics and surfactants, but also resistant to physical stress[14,15].

LPS has a tripartite structure, made of lipid A, the core oligosaccharide, and the O-antigen. Lipid A is the hydrophobic moiety, comprising 4 to 7 acyl tails attached to a disaccharide of two phosphorylated N-acetyl glucosamine units. The headgroup of lipid A is decorated with a core oligosaccharide of approximately 10 sugars, which can be further extended with the addition of the O-antigen, a highly variable sugar segment composed of repeated oligosaccharide units[16–18]. The most conserved portion of LPS is Kdo-lipid A (KLA) consisting of lipid A and two 3-deoxy-D-manno-oct-2-ulosonic acid (Kdo) sugars from the oligosaccharide core[19]. The remaining saccharide components are generally dispensable in *Escherichia coli* under laboratory growth conditions. Anterograde LPS transport (Lpt) from its site of biosynthesis at the inner membrane (IM) to the OM is mediated by the 7-component Lpt pathway, which forms a transenvelope bridge[7,20]. An ATP-binding cassette (ABC) transporter, $LptB_2FG$, extracts LPS from the IM and transfers it to the IM-anchored protein LptC. LptA then ferries LPS across the periplasm, from LptC to the OM translocon LptDE. During transport, the lipid tails of LPS move along the hydrophobic grooves formed by stacked β-taco domains[21], including those of LptFG and LptC at the IM, LptA in the periplasm, and LptD at the OM[22,23] (Fig. 1a).

The transmembrane domain of LptD forms a β-barrel composed of 26 β-strands (designated β1 to β26), with an internal lumen partially occluded by the lipoprotein LptE[24–27]. The β-barrel domain of LptD is folded around LptE at the BAM complex[28]. Covalent bonds connect the β-taco domain to both the N- and C-terminal portions of the β-barrel domain[29–31]. The polypeptide backbone linking the β-taco to β1 and the double-disulfide bond anchoring the β-taco to the periplasmic turn β24-β25 create a conduit from the periplasm to the OM. It has been proposed that the acyl tails of LPS enter the membrane near the LptD β-taco/β-barrel interface region, whereas the LPS polysaccharides transverse the β-barrel lumen adjacent to LptE and exits at the cell surface through a putative lateral gate between the terminal β-barrel strands, β1 and β26[24,25,32,33] (Fig. 1a). Experimental evidence supports possible opening of the LptD lateral gate in agreement with this model[25,34,35]. Nevertheless, all high-resolution structures have captured the LptD lateral gate in a closed conformation. Complete separation of the lateral gate strands has been observed only upon molecular dynamics (MD) simulations under conditions of high negative pressure[25] or upon binding with macrobodies[35]. Consequently, the precise mechanism by which LPS is selectively released into the OM external leaflet remains unclear.

We recently identified the lipoprotein LptM as a stoichiometric component of the OM LPS translocon[36]. Our findings demonstrated that, like LptE, LptM associates with LptD during the folding of its transmembrane domain at the BAM complex, a step that facilitates subsequent disulfide bond formation between the β-barrel and β-taco domains[36]. However, whether LptM plays a role in LPS translocation has remained unknown.

In this work, during the determination of the LptDEM structure, we identified and characterized an additional component of the translocon, the lipoprotein YedD, which we have renamed LptY in accordance to the Lpt nomenclature[37]. Most importantly, functional assays and structural analyses of the translocon holo-complex LptDEMY and of its subcomplexes, LptDEM, LptDEY and LptDE, reveal that LptD operates through a conformational switch between contracted and extended states of its β-taco/β-barrel interface near the β1-β26 lateral gate. Whereas LptY stabilizes the LptD β-taco domain,

LptM is positioned at the lateral gate facilitating its opening and access by LPS.

## Results

### The OM LPS translocon interacts with LptY (formerly YedD)

To investigate whether LptM plays a role in LPS translocation, we aimed at determining the structures of LptDE translocon with and without LptM. We purified both LptDEM and the core translocon LptDE via nickel-affinity chromatography and gel filtration. As baits, we used respectively LptM[His] expressed in wild-type cells or LptE[His] expressed in Δ*lptM* cells (Fig. 1b). Notably, both eluates contained an additional protein migrating at an apparent molecular mass of approximately 15 kDa. Bottom-up mass spectrometry on the corresponding gel bands identified this protein as LptY, an OM lipoprotein of ~14 kDa[38] (Supplementary Fig. 1a, b, see also Supplementary Methods). Phylogenetic analysis showed that, like *lptM*, *lptY* is conserved within *Enterobacteriaceae* (Supplementary Fig. 1c) and *Erwiniaceae*, except for species presenting highly reduced genomes, which often lack also *lptD and lptE*. Furthermore, *lptY* is present in some *Pectobacteriaceae*, but is absent in other families of *Enterobacterales* such as *Yersiniaceae*, *Hafniaceae*, *Morganellaceae* and *Budviciaceae*, which retain *lptM* along with *lptD* and *lptE* (Supplementary Figs. 2 and 3, see also Supplementary Methods).

In our previous study, native mass-spectrometry analysis of LptDEM and LptDE revealed respectively 1:1:1 and 1:1 stoichiometries[36]. Revisiting these spectra in consideration of LptY's identification, we detected a subpopulation of both LptDEM and LptDE with a mass shift of 14,301 Da, consistent with complexes bound to LptY. This yielded LptY-bound complexes with a 1:1:1:1 and a 1:1:1 stoichiometries (Supplementary Fig. 4; see also in ref. 36 and the PRIDE repository dataset PXD068376). MS spectra obtained in the low m/z region identified a species at 14,302 Da, corresponding to LptY dissociated from LptDEM[His] and LptDE[His] complexes (Supplementary Fig. 4b, d). Supporting these findings, LptY pull-down experiments co-isolated LptD, LptE, and LptM, indicating stable association with the LPS translocon (Supplementary Fig. 5). Thus, LptY emerges as a component of the OM LPS translocon capable of binding LptDE alongside LptM. We refer to LptDE bound to both LptM and LptY as the LPS translocon "holo-complex".

### Structural determination of the LPS translocon holo-complex and sub-complexes

Using cryo-EM and single particle analysis, we determined the structures of the purified OM LPS translocon complexes in n-dodecyl-β-D-maltopyranoside (DDM) micelles (Fig. 1c–f and Supplementary Figs. 6, 7 and 8). The structure of the core complex, LptDE, was solved at an overall resolution of 2.74 Å for the transmembrane portion including the LptD β-barrel and LptE. However, in the reconstructed maps, no density was visible for the LptD β-taco domain (Fig. 1f and Supplementary Fig. 8d), indicating conformational flexibility of this domain as previously documented[24,34]. Flexibility of the β-taco domain in the absence of LptM was also suggested by our previous hydrogen-deuterium exchange (HDX)-MS results, showing high deuteration levels for the β-taco in the LptDE core complex[36]. Furthermore, we solved the structures of LptDE bound to either LptM, LptY, or both proteins (holo-complex) at resolutions of 2.47, 2.62, and 2.63 Å, respectively. In these complexes, binding of LptM, LptY, or both stabilized the β-taco domain, enabling its structural determination (Fig. 1c–e and Supplementary Fig. 8a–c).

### LptM intercalates at the interface between the LptD β-taco domain and the β-barrel lateral gate

The electron density map allowed de novo model building of the N-terminal portion of LptM, including the first 10 amino acid residues of

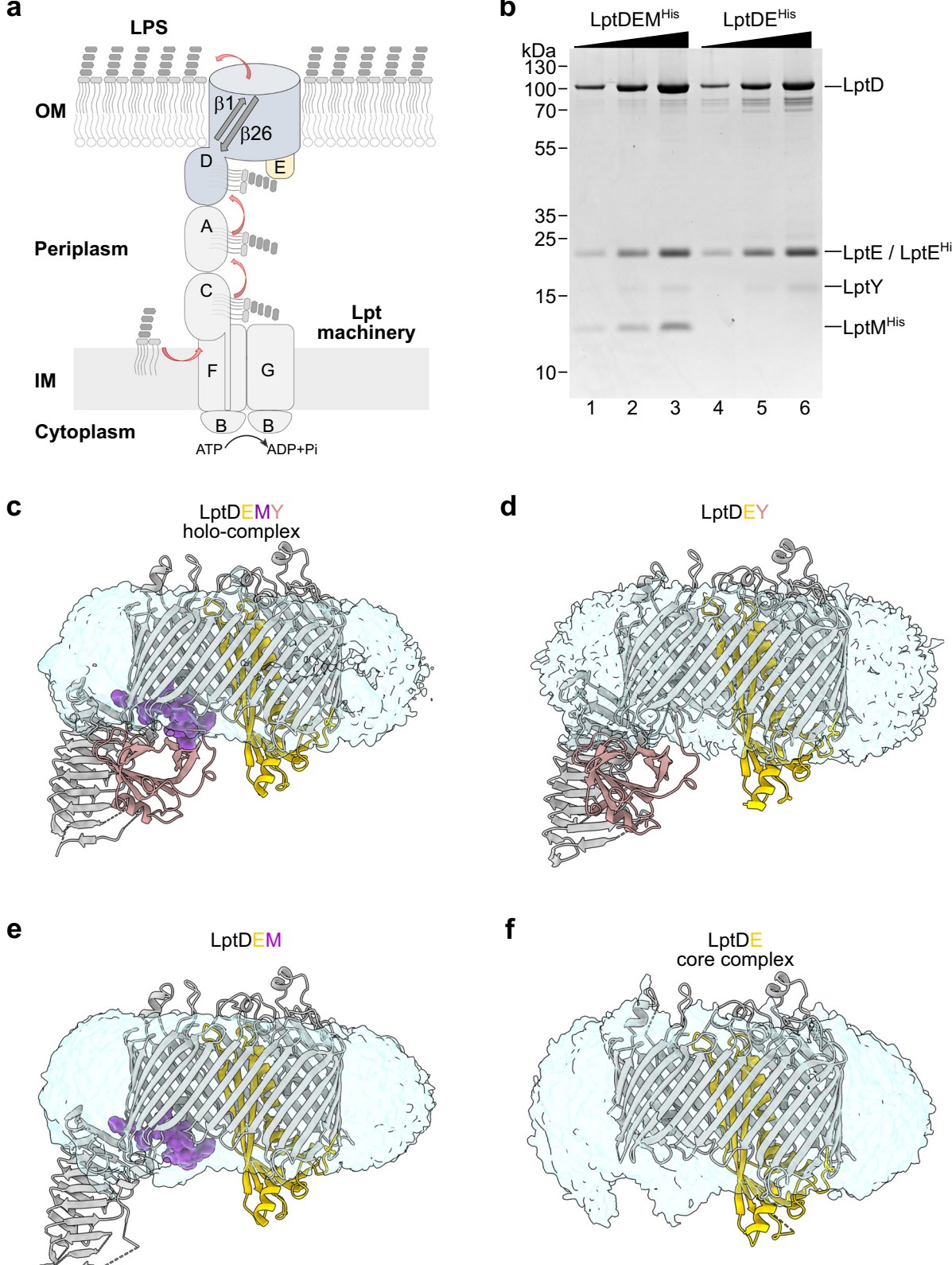

**Fig. 1 | Purification and structural determination of OM LPS translocon complexes. a** Schematic representation of the Lpt pathway. **b** Incremental amounts of the indicated purified translocons were loaded on SDS-PAGE and revealed by Coomassie Blue staining. LptDEM^His was purified from wild-type cells transformed with pLptDEM^His whereas LptDE^His from Δ*lptM* cells transformed with pLptDE^His. The result is representative of three independent experimental repeats. **c**–**f** Cryo-EM

structures of the LptDEMY translocon holo-complex or the indicated sub-complexes in DDM micelles. LptD (gray), LptE (yellow), and LptY (pink) are shown in ribbon representation, whereas LptM (purple) is shown in sphere representation. The transparent blue surfaces represent the DDM micelle environment surrounding the protein complexes.

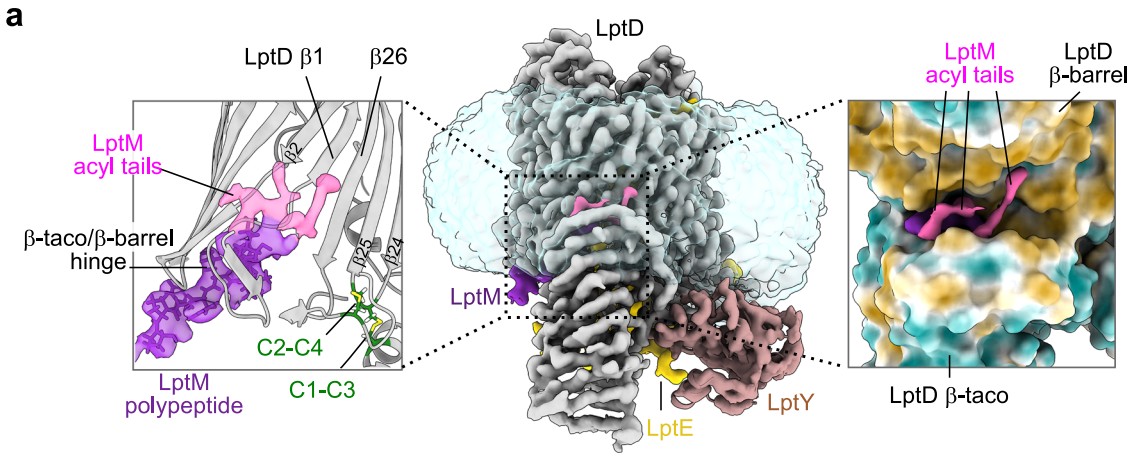

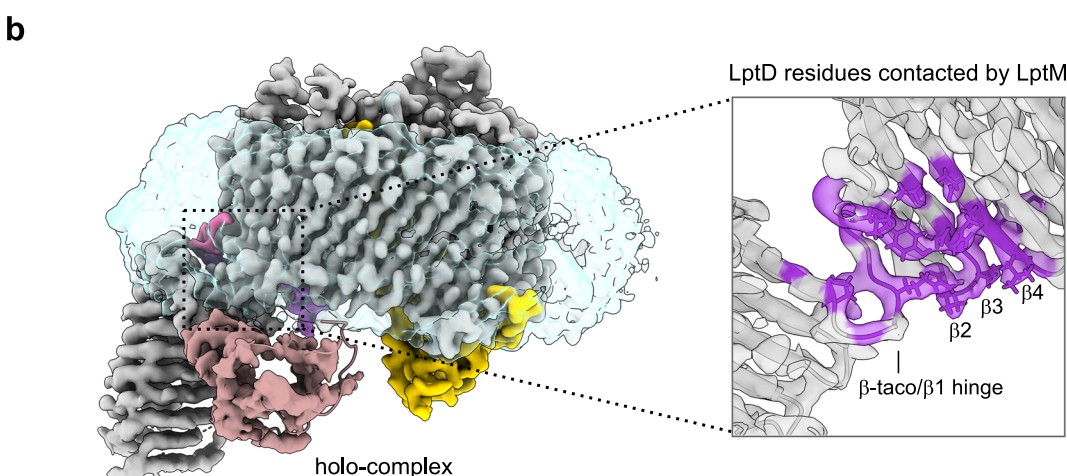

**Fig. 2 | Interaction of LptM with the LptD β-barrel/β-taco interface. a** Left, Zoom on LptM in the structure of LptDEMY, highlighting the electron density corresponding to the acyl tails of LptM (pink) and the electron density corresponding to the LptM N-terminal moiety (LptM$_{20-29}$, purple) situated near the lateral gate of LptD (β1-β26). The β-taco/β-barrel hinge and the LptD inter-domain disulfide bonds surrounding LptM are indicated. LptD Cys31 (C1), Cys173 (C2), Cys724 (C3) and Cys725 (C4) are shown in green with disulfide bonds in yellow. Central, cryo-EM map of the LptDEMY complex colored according to the protein chains (LptD in gray, LptE in yellow, LptM in purple, LptY in pink). Right, Zoom on the acyl tails of LptM and the surrounding surface of LptD colored based on hydrophobicity of the amino acid side chains (brown, hydrophobic; cyan, hydrophilic). **b** Internal section of the LptD β-taco/β-barrel hinge region in the holo-complex. The LptD residues in contact with LptM are shown in purple. Cryo-EM density maps are displayed at threshold density of 0.09.

the mature sequence following its N-terminal Cys residue (LptM$_{20-29}$) (Fig. 2a). This segment is part of the Pfam motif PF13627, which is highly conserved across bacterial species harboring *lptM*[36]. The LptM N-terminal region alone (LptM$^{Δ44-67}$) is sufficient to mediate its co-purification with the translocon (Supplementary Fig. 9a). Conversely, an internal truncation of residues 22–30 of LptM (which includes a significant portion of the N-terminal part resolved in our cryo-EM structure, LptM$^{Δ22-30}$) strongly impairs association with the translocon (Supplementary Fig. 9a). The structurally resolved portion of LptM forms extensive contacts with the internal surface of the LptD β-barrel, particularly with residues located at the periplasmic-facing side of β1-β4, as well as with the β-taco/β-barrel hinge region preceding β1 (Fig. 2b). This interaction between LptD and LptM remains largely unchanged in the absence of LptY (Supplementary Fig. 9b, c; Supplementary Table 1). In both LptDEM and LptDEMY structures, we resolved portions of the LptM lipid tails in positions proximal to the external surface of the LptD lateral gate. Notably, one of the acyl tails intercalates between the outward-facing side chains of β1 and β26 (Fig. 2a and Supplementary Fig. 9b). During the revision of our study, two new published papers reported on the structures of the *E. coli* LptDEM complex[39] and the *Pseudomonas aeruginosa* LptDE complex heterologously expressed in

*E. coli* and consequently bound to *E. coli* LptM[40]. Both studies resolved only the first ten amino acid residues of LptM, showing its N-terminal cysteine residue intercalating the LptD β-barrel lateral gate, and the downstream LptM portion in the β-barrel lumen making contacts with LptD β1-β4, as observed in our structures. These findings corroborate the notion that LptM is a stable component of the OM LPS translocon.

The LptD β-taco domain is anchored to both sides of the β-barrel lateral gate. On the N-terminal side, the polypeptide backbone connects the β-taco to β1. On the C-terminal side Cys31 (C1) and Cys173 (C2) in the β-taco are linked via disulfide bonds respectively to Cys724 (C3) and Cys725 (C4) located in the β24-β25 periplasmic turn (Fig. 2a)[29]. Positioned at the β1-β26 lateral gate, LptM is constrained between the two interdomain connections and the inner leaflet of the membrane, thus raising the question of how LptM reaches its location in the translocon. We had previously shown that LptM is assembled with LptD at the BAM complex[36], where an early oxidation intermediate of LptD lacking interdomain disulfides undergoes folding[30]. It is therefore likely that the N-terminus of LptM is positioned within the translocon before the β-taco and the β-barrel domains are connected by disulfide bonds[36].

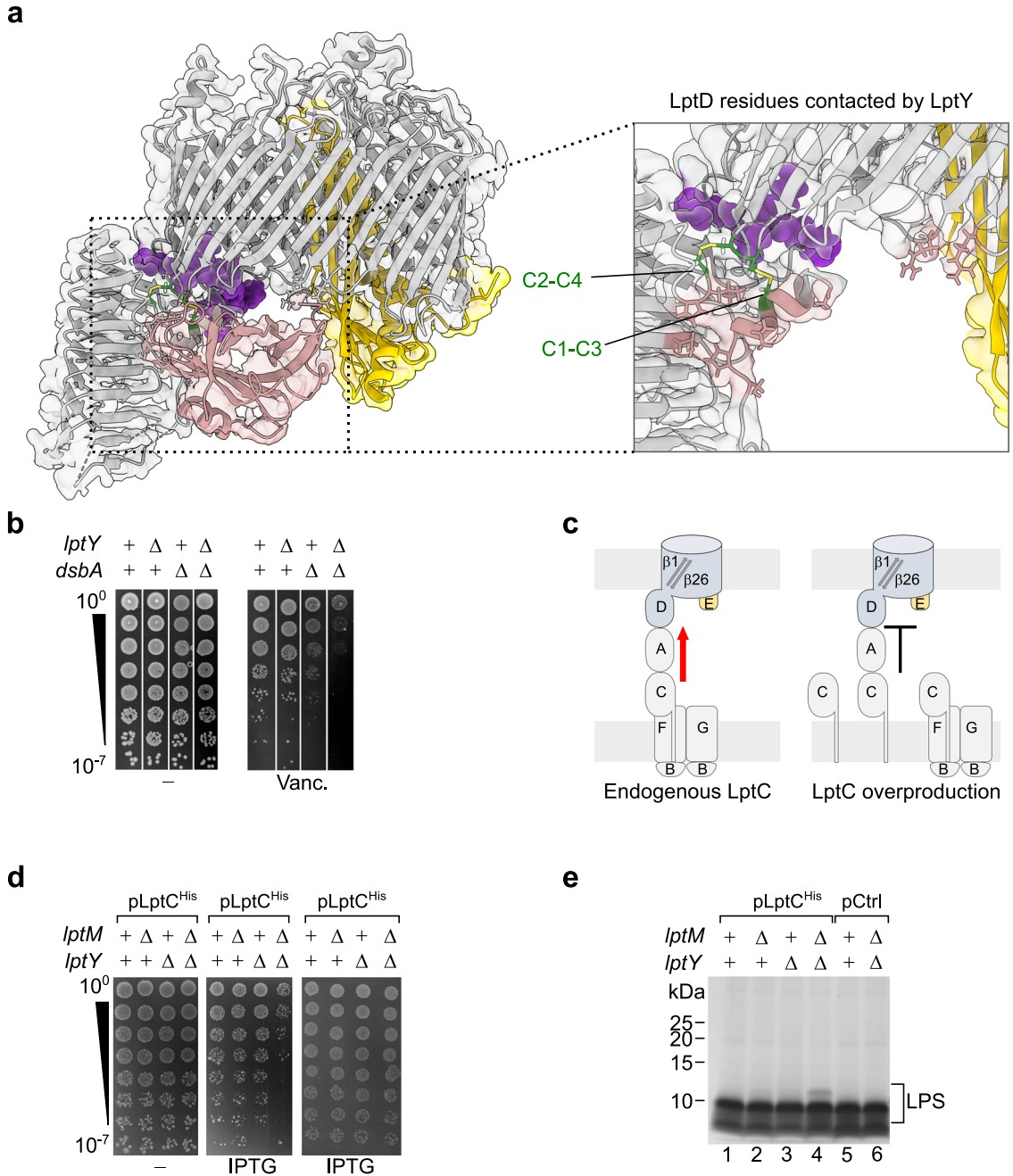

**Fig. 3 | Interaction of LptY with the LptD β-taco domain. a** Zoom on the LptD area in contact with LptY, proximal to the LptD inter-domain disulfide bonds. Proteins shown in ribbon representation are LptD (gray), LptE (yellow), and LptY (pink). LptM is shown in sphere representation (purple). **b** Drop dilution test of the indicated strains on regular LB agar media or LB agar supplemented with vancomycin. **c** Schematics of the predicted effects of LptC overproduction on Lpt bridges. **d** drop dilution spot test of the indicated strains on MacConkey agar media or on regular LB agar media, as indicated. **e** SDS-PAGE migration and silver staining of LPS extracted from strains tested in (d) grown on LB media supplemented with IPTG. The result is representative of three independent experimental repeats.

## LptY interacts with LptD β-taco domain independently of LptM

The cryo-EM density map revealed clear features corresponding to LptY adjacent to the LptD β-taco domain. We generated a predicted structure of *E. coli* LptY using AlphaFold 2[41], which was consistent with the previously determined X-ray crystal structure of the non-lipidated variant of its ortholog from *Klebsiella pneumoniae* (PDB: 4HWM). The predicted AlphaFold 2 model could be fitted into the experimental density using rigid body docking, despite its limited resolution in this region of the map (Fig. 3a and Supplementary Fig. 10a). LptY adopts an 8-stranded β-barrel fold flanked by a short N-terminal 3₁₀-helix and a C-terminal α-helix. This organization of structural motifs resembles

that of lipocalins, a family of proteins capable of interacting with hydrophobic ligands[42]. Notably, LptY contains two conserved cysteine residues (Supplementary Fig. 1c), one in β-strand 3 and one in the C-terminal α-helix, which are positioned at a distance compatible with disulfide bond formation (Supplementary Fig. 10a). Structural analysis of the two LptY-containing complexes revealed similar contact areas with the external surface of LptD β-taco domain and its N-terminal α-helix (Fig. 3a and Supplementary Table 2). During our study, another paper has described the cryo-EM structure of the *E. coli* LPS translocon bound to LptY, showing a similar interaction between the β-taco domain of LptD and LptY[43]. The deposited data from this study further

include an unassigned extra density located at a position similar to that occupied by LptM in our structure (Supplementary Fig. 9d)[43]. Our pull-down of LptY[His] from cells co-expressing endogenous LptD and a N-terminally truncated version lacking the β-taco domain, LptD[β-barrel], only co-eluted full-length LptD, thus proving the critical role of the β-taco domain for LptY binding. In contrast, pull-down of LptM[His], which interacts with the LptD β-barrel domain, co-eluted both the full-length and the N-terminally truncated β-barrel forms of LptD (Supplementary Fig. 10b)[36]. The independent binding of LptD to LptM and LptY is further highlighted by the fact that we could determine the structures of the core translocon LptDE bound to either one of the herein identified translocon lipoproteins (Fig. 1d, e).

Given LptY's interaction proximal to a region of LptD that undergoes oxidative maturation, we investigated whether LptY plays a role in LptD biogenesis. Deletion of *lptY* does not affect the levels of oxidized LptD (Supplementary Fig. 10c), nor leads to accumulation of LptDE at the BAM complex (Supplementary Fig. 10d). Interestingly, we found LptY to be critical for OM integrity against the large-molecular weight antibiotic vancomycin in cells lacking the cysteine oxidase *dsbA* (Fig. 3b). DsbA catalyzes LptD oxidation[29], and cells lacking this enzyme show reduced LptD levels associated with mild sensitivity to vancomycin[36]. Conversely, Δ*lptY* cells were not sensitive to the tested antibiotic concentration. However, deleting *lptY* in Δ*dsbA* resulted in a synergistic detrimental effect that further enhanced OM permeability compared to Δ*dsbA* alone (Fig. 3b), highlighting the importance of LptY when LptD levels are reduced.

Similar to LptM, which we showed to associate with IM and periplasmic Lpt components[36], LptY purification co-eluted periplasmic LptA (Supplementary Fig. 5), indicating that both LptM and LptY are part of transenvelope Lpt bridges. Because our structures showed that LptM and LptY stabilize the LptD β-taco domain, which connects the OM and IM portions of the Lpt bridge, we investigated whether cells lacking LptM and LptY would be more sensitive to perturbations of Lpt bridges. To explore this hypothesis, we examined the effects of over-producing LptC. This condition was suggested to titrate the OM translocon via LptA, as an excess of LptC relative to LptB$_2$FG contributes to form incomplete bridges[22]. We reasoned that reducing the number of complete Lpt bridges would make cells more dependent on efficient functioning of the remaining IM-OM Lpt connections (Fig. 3c, diagram). Upon LptC overproduction, cell susceptibility to bile salts (MacConkey media) increased, especially in a strain harboring both deletions of *lptM* and *lptY* (Fig. 3d). The enhanced susceptibility of the *lptM* and *lptY* double deletion strain correlates with a modification of a fraction of LPS extracted from its envelope, which retards its migration on SDS-PAGE (Fig. 3e). This modification might be due to decoration with colanic acid of LPS that accumulates at the IM because of inefficient transport along the Lpt bridges[7,36,44]. Taken together, these results suggest that LptM and LptY are OM LPS translocon components that contribute to optimal functioning of Lpt bridges.

## LptD functions through the contraction and extension of its β-taco/β-barrel interface

Our structural analysis has revealed that both LptM and LptY contribute to stabilizing the LptD β-taco domain, enabling its structural determination. To explore other potential variations within the translocon holo-complex, its variants lacking either LptY or LptM, and the core complex, we superimposed the four complexes and assessed the root mean square deviation (RMSD) of atomic distances. The membrane portion of the core translocon was superimposable with that of the holo-complex, resulting in an RMSD lower than 0.1 Å for the largest portion of LptE and the LptD β-barrel domain (Supplementary Fig. 11a, d). Similarly, the holo-complex and the LptDEM complex showed an RMSD lower than 0.1 Å for the membrane-embedded portion of the translocon and an RMSD ranging from 0.1 to 0.8 Å for the β-taco domain (Supplementary Fig. 11a, b).

Remarkably, superimposition of the holo-complex with the variant lacking LptM (LptDEY) revealed significant deviations in the β-barrel lobe containing the lateral gate, with >2 Å deviations for β1-β4 and β26 (Supplementary Fig. 11a, c). These conformational variations in the LptD β-barrel domain result from a +2 increment in the β-barrel shear number[45]. This means that β1 slides along β26 of two amino acid positions towards the periplasm (Fig. 4a). Simultaneously, the hinge loop at the N-terminal side of β1 bends toward the β-barrel lumen, causing the β-taco to move toward the membrane (Fig. 4a and Supplementary Fig. 12a). The convergence of the β-taco and β1 movement reduces the gap between the two LptD domains (measured between residues P219 in the β-taco and N232, the first residue of β1) from 19.7 Å in presence of LptM to 12.0 Å in its absence (Fig. 4b). Henceforth we will name these distinct LptD conformations "contracted" state, as observed in the structure of LptDEY, and "extended" state, as observed in LptDEMY and LptDEM structures. The structural deviations observed at the β-taco/β-barrel interface in the absence of LptM fit well with the bimodal distribution of hydrogen-deuterium exchange (HDX) kinetics that we have previously documented using HDX-mass spectrometry on the same sample. LptM, instead, abrogates this bimodal behavior, stabilizing the interface between the two domains[36].

## Pivots in the contracted-extended state transition

Whereas high dynamics of the β-taco domains are expected, with different orientations relative to the membrane plane having been previously reported[24,34,46], the major structural variations observed in the β-barrel domain of LptD are unprecedented. Two conserved proline residues important for function[24], P231 and P246 respectively in LptD β1 and β2, act as pivots for the contracted-to-extended state transitions. P231 serves as a re-orientation point for the luminal loop connecting the β-taco and β1 (Fig. 4b). In the extended state, a portion of this loop prolongs the region of H-bonding between β1 and β2, whereas in the contracted state the loop bends toward the interior of the barrel weakening the β2-β1 pairing (Fig. 4b, c). The other conserved proline, P246, is located where β2 flexes, reducing its tilt angle with the membrane plane at the inner leaflet (Fig. 4b and Supplementary Fig. 12a, Right). This flex point compensates for the increased β-barrel shear number, maintaining the periplasm-oriented hydrophobic belt of the β-barrel within the membrane plane.

Given the identification of these two distinct conformational states of the *E. coli* LPS translocon, we compared our structures to those of LptDE previously determined by X-ray crystallography. We specifically chose LptDE from *K. pneumoniae* (PDB 5IV9)[24] and *Shigella flexneri* (PDB 4Q35)[26], as these bacteria are closely related to *E. coli*, with a high degree of amino acid sequence identity among their LptD orthologues[24]. Furthermore, in both cases the structure of full-length LptD was resolved. Our comparison revealed that the pairing pattern of β1-β26 in the extended conformation (LptDEM and LptDEMY) superimposes well with that observed in the LptD structures of *K. pneumoniae* and *S. flexneri* (Fig. 4d and Supplementary Fig. 12b). Furthermore, the recently determined structure of the *E. coli* and *P. aeruginosa* LPS translocon complexes showed the same β1-β26 pairing pattern observed for *K. pneumoniae* and *S. flexneri* LptD[39,40,43]. Hence the contracted state of LptD identified in our study represents an unprecedented conformation of the LPS translocon.

## Lateral opening of LptD in the contracted state is crucial for function

Our structural analysis uncovered the LptD contracted state in samples lacking LptM. To test whether the LptD contracted state represents a physiological conformation in cells expressing LptM, we employed a cysteine crosslinking approach. A previous study on *Salmonella typhimurium* LptD showed that locking the β1-β26 pairing via interstrand disulfide bond formation is lethal, supporting a mechanistic model where lateral gate opening is essential for LptD function[25].

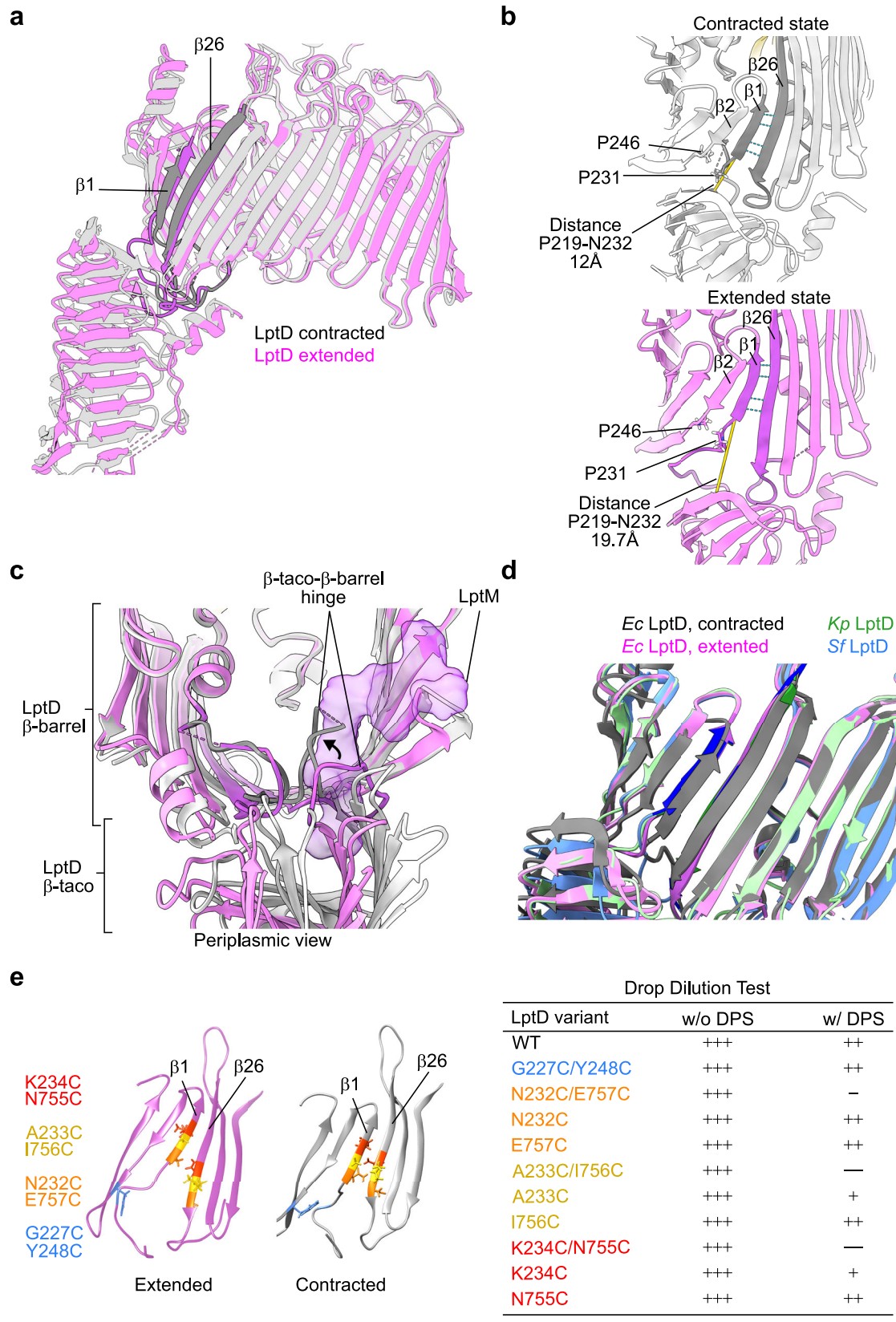

**e**

| Drop Dilution Test | | |
|---|---|---|
| LptD variant | w/o DPS | w/ DPS |
| WT | +++ | ++ |
| G227C/Y248C | +++ | ++ |
| N232C/E757C | +++ | − |
| N232C | +++ | ++ |
| E757C | +++ | ++ |
| A233C/I756C | +++ | — |
| A233C | +++ | + |
| I756C | +++ | ++ |
| K234C/N755C | +++ | — |
| K234C | +++ | + |
| N755C | +++ | ++ |

Following this strategy, we tested the effects of forming disulfide bonds between β1 and β26 paired as in the LptD contracted states. As a control experiment, we also tested the effect of forming a disulfide bond between β1 and β2 as well as the effects of introducing the single cysteine residues at the lateral gate. We generated a diploid *E. coli* strain co-expressing endogenous LptD and a plasmid borne LptD

variant with the additional cysteine residues. We then deleted endogenous (wild-type) LptD via P1 phage transduction, and assessed functionality of the mutant LptD variants by testing the viability of the obtained clones on regular media or media containing the oxidant 4,4-dipyridyl disulfide (4-DPS) to facilitates cysteine oxidation (Fig. 4e and Supplementary Fig. 13a). The LptD variants containing the cysteine

**Fig. 4 | Extended and contracted conformational states of LptD.**
**a** Superimposition of LptD conformations observed in the structure of LptDEMY (purple) or LptDEY (gray). The terminal β-barrel strands β1 and β26 are shown in darker colors. Both proteins are shown in ribbon representation. **b** Zoom on the LptD β-taco/β-barrel interface of LptD in the contracted state (as in the structure of LptDEY) or LptD in the extended state (as in the structure of LptDEMY). The side chains of the pivot residues P246 and P231 are shown. The distance between P219 and N232 is indicated in yellow. **c** Periplasmic view of the β-taco/β-barrel hinge loop (purple in LptDEMY and gray in the LptDEY) that reorients in the absence of LptM (shown as purple surface). **d** Superimposition of the structure of LptD in the extended (purple) or contracted (gray) states with the structures of LptD from *Klebsiella pneumoniae* (green) or *Shigella flexneri* (blue). **e** Left, Strands β1-β3 and β24-β26 of the LptD β-barrel domains are shown in ribbon representation, purple in the case of extended LptD (structure of LptDEMY) and gray in the case of contracted LptD (structure of LptDEY). The side chains of the residue pairs replaced with cysteines are shown in different colors. Right, Viability of strains expressing different LptD variants in presence or absence of 4-DPS as indicated. The table summarizes the results of the drop test illustrated in Supplementary Fig. 12a. +++, presence of $10^{-5}$ dilution spots; ++, $10^{-4}$ dilution spots; +, $10^{-3}$ dilution spots; −, $10^{-2}$ dilution spots; −−, $10^{-1}$ dilution spots.

pairs N232C/E757C, A233C/I756C, or K234C/N755C, each forming disulfide bonds between β1 and β26 as arranged in the contracted state, supported wild-type-like growth on regular media but caused marked growth defects in the presence of 4-DPS. This growth defect appeared specific to the β1-β26 cysteine pairs, as single cysteine substitutions in β1 or β26 still supported growth. These results suggest that, upon 4-DPS treatment, the locking of the LptD lateral gate strongly inhibits growth (Fig. 4e and Supplementary Fig. 13a). Importantly, in the absence of the oxidant (that is under growth permissive conditions), the cysteine pairs in β1 and β26 did not interfere with the release of LptD from the BAM complex, nor affected the interaction of LptD with LptM (Supplementary Fig. 13b), even though they somewhat impacted the formation of mature LptD$^{Ox}$ to different degrees (Supplementary Fig. 13c). In the control experiment, the LptD variant containing the cysteine pair in β1 and β2 (G227C/Y248C) promoted cell survival also in the presence of the oxidant (Fig. 4e and Supplementary Fig. 13a). Taken together, these results suggest that in cells expressing LptM, the lateral gate of LptD can be locked in a contracted conformation by an inter-strand disulfide bond between β1-β26, thereby inactivating LptD. Hence, the contracted form of LptD appears to represent a physiological translocon state, in which lateral gate opening is crucial for function.

## LptM induces gate dynamics, facilitating β1-β26 strand separation

Given the structural differences at the lateral gate between the two LptD states, we questioned whether there was sufficient space to model LptM with LptD in the contracted state. To explore this, we superimposed both conformations and transferred the coordinates of LptM from the extended state to the LptY-bound contracted state of LptDE. This transfer resulted in minimal atomic clashes, which could be further resolved with energy minimization, using the Gromacs MD simulation package[47] (see dataset deposited at the Zenodo repository, [https://doi.org/10.5281/zenodo.16643358], see also Supplementary Fig. 14c, g, i). To further assess the impact of LptM and LptY on the core complex, we next performed molecular dynamics simulations of the OM translocon embedded in asymmetric lipid bilayer containing LPS in the external leaflet. The simulations were initiated with LptD in either the extended (LptDEM) or in the contracted (LptDEY) states allowing us to evaluate the contributions of LptM and LptY to translocon lateral gate dynamics (see dataset, [https://doi.org/10.5281/zenodo.16643358]; Supplementary Figs. 14 and 15).

We focused our analysis on the pairing of β1-β26 and β2-β1, which vary between the two LptD states. In fact, whereas β1-β26 present alternative pairing patterns, β2-β1 present a number of H-bonds that is higher in the extended conformation as the hinge segment preceding β1 aligns parallel to β2 (Fig. 4a, b and Supplementary Fig. 16a). The presence of LptM partially reduced β1-β26 H-bonding, from an average of $3.31 \pm 0.68$ in LptDE and $3.38 \pm 0.74$ in LptDEY to $2.63 \pm 0.83$ in LptDEM and $2.79 \pm 0.78$ in LptDEMY. Interestingly, the addition of KLA built in the β-taco domain of the holo-complex further decreased β1-β26 pairing, averaging only $2.01 \pm 0.46$ H-bonds (Supplementary Fig. 16a). Strikingly, when the simulations were initiated with LptD in

the contracted state, larger fluctuations in the number of H-bonds were observed not only between β1 and β26 but also between β2 and β1, with both pairs presenting a reduced average of H-bonds in the presence of LptM (Supplementary Fig. 16a). In all cases, β-strand separation was more prominent on their periplasmic side (Supplementary Fig. 16b). Remarkably, full separation of β1-β26 could be observed for a discrete period at least in one of our simulation repeats (Fig. 5b and Supplementary Movie). Also in the contracted state, the addition of KLA consolidated the weakening of β-strand pairing (Supplementary Fig. 16a). Taken together, these simulation results indicate that LptM enhances gate dynamics, with this effect being also promoted by the addition of KLA into the β-taco domain. In contrast, LptY had minimal effects on LptD lateral gate dynamics (Supplementary Fig. 16a, b). In all cases, the structures do not switch between extended and contracted states and vice versa, at least over the 500 ns course of our MD simulations setup.

## LptM operates at the translocon during LPS gating

Given its positioning at the β-taco/β-barrel interface and its effect on lateral gate dynamics, we investigated whether LptM influences the interaction between distinct domains of LptD and LPS. LptM is required for the proper oxidative maturation of LptD. Nevertheless, upon overproduction of LptDE in Δ*lptM* cells, we showed that the majority of LptD (approximately 70%), is correctly oxidized. This fraction increases to 95% when LptM is overproduced together with LptD and LptE[36]. Despite this marginal difference in LptD oxidation, we probed distinct LptD domains to monitor their interactions with LPS. We introduced the photoactivable amino acid analogue para-benzoyl phenylalanine (pBpa) at specific positions within the β-taco domain, the hinge region and β1 (Fig. 5). We aimed to maximize the probability of detecting interactions between LPS and translocon intermediates by replacing with pBpa the amino acids whose side chains project into the hydrophobic cavity of the β-taco or toward the internal lumen of the β-barrel. Upon UV irradiation and translocon purification, we detected crosslinks between LptD and LPS along the β-taco domain and β1 (Fig. 5d). Notably, positions T236 at the end of β1 and N239 in the β1-β2 loop showed the highest crosslinking efficiency. The side chains of T236 and N239 remain oriented toward the interior of the barrel during our MD simulations, although that of T236 showed a higher degree of dihedral angle variation (Supplementary Fig. 17a, b). The efficient interaction of these amino acid positions with LPS indicates that the apical region of β1, which repositions during the contracted-extended conformational switch observed in our structural analyses, represents a major LPS binding site. Positions near the β-taco/β-barrel interface interact with LPS with lower efficiency, and these were also found to bind LptM, consistent with our structural analysis. Crosslinking to LPS within the β-taco domain occurred with a similar efficiency regardless of whether LptM was produced. Remarkably, at position T236, which is in β1 exposed to the lateral gate slit, the efficiency of LPS-crosslinking was drastically reduced by >90% in the absence of LptM (Fig. 5d, e). Taken together, these results suggest that LptM influences the mature, active translocon, facilitating LPS access to a position within the β-barrel lateral gate. This observation led us to

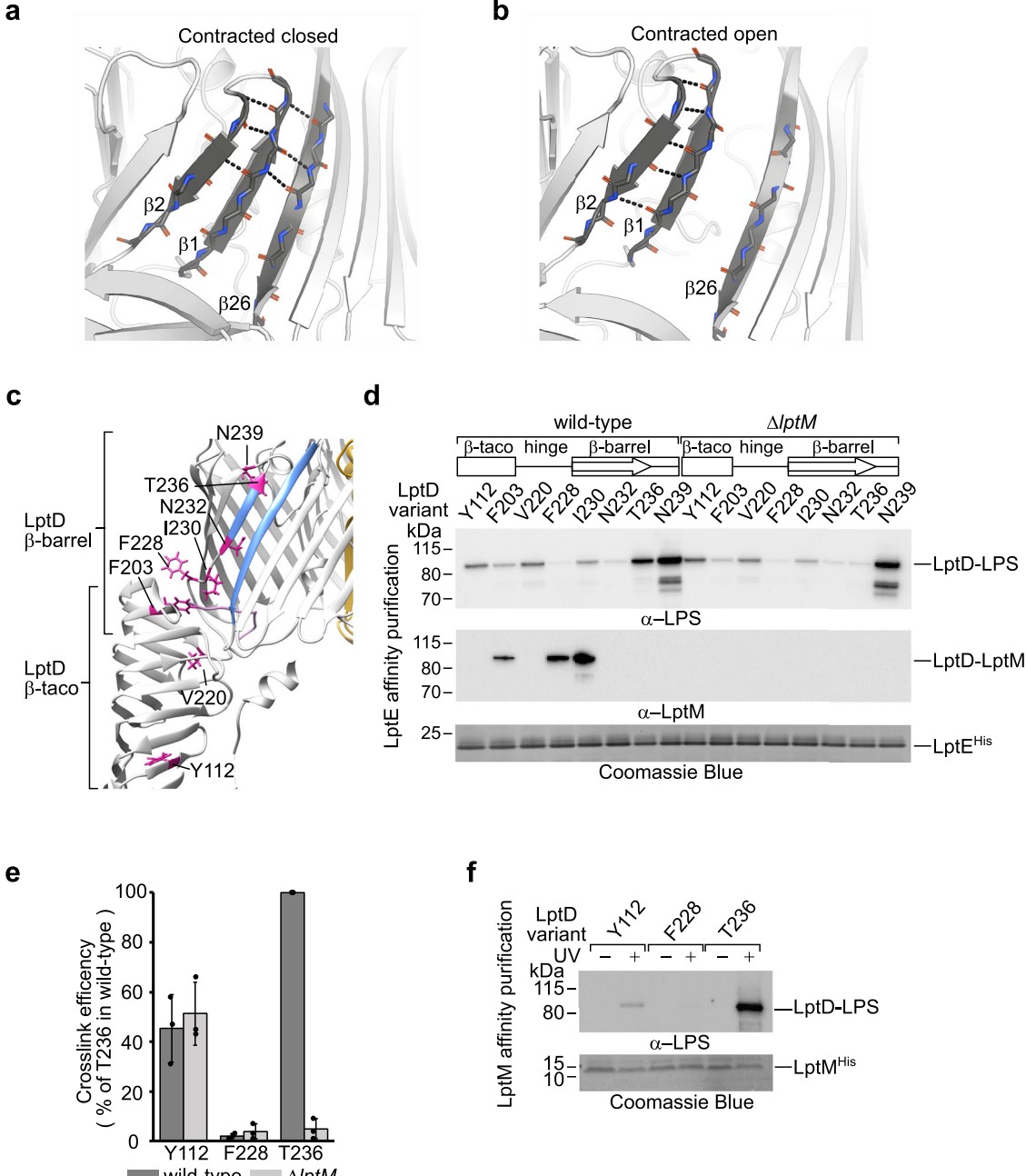

**Fig. 5 | LptM enhances LptD lateral gate dynamics. a** Snapshots of the LptD lateral gate conformation in LptDEY containing docked LptM at the start of MD simulations. For clarity, only LptD is shown. β1 and β26 form multiple hydrogen bonds. **b** Snapshots of the LptD conformation in LptDEY containing docked LptM after 140 ns of MD simulations in one of the three independent runs (Run 1, see Supplementary Movie). β1 and β26 are fully separated. In **a** and **b**, β2, β1, and β26 are shown in darker gray. **c** The structure of LptDEM (extended state) is shown in ribbon representation to highlight the side chains (purple) of LptD amino acids replaced with pBpa for site-directed photocrosslinking. **d** UV induced photocrosslinking of the indicated strains transformed with pLptDE^His harboring amber mutations at the indicated amino acid positions of LptD. Upon in vivo

photocrosslinking, the envelope fractions were solubilized with DDM and subjected to nickel-affinity purification of LptE^His. **e** Quantitation of ECL signals of LptD-LPS crosslinking adducts for the indicated LptD positions in cells expressing or lacking LptM, as shown in (**d**). Data are represented as means, ± s.e.m. (*n* = 3 biological replicates). Values are normalized to the intensity of the crosslink adduct obtained for LptD containing pBpa at position T236 and expressed in wild-type cells. Source Data are provided as a Source Data file. **f** UV induced photocrosslinking of the indicated strains transformed with pLptDEM^His harboring amber mutations at the indicated positions in LptD. Solubilized membranes were prepared as in (**d**), and subjected to nickel-affinity purification of LptM^His. The result is representative of three independent experimental repeats.

explore whether LptM plays a direct role at the active translocon during LPS secretion. Using a similar site-directed crosslinking approach, we selected three positions within LptD to monitor interactions with LPS by LptM^His pull-down, thereby assessing for the simultaneous binding of LPS and LptM to the translocon. Our analysis clearly shows that LPS binds to position 236 of LptD without requiring

the release of LptM from the translocon (Fig. 5f). Notably, in both the contracted and the extended states of LptD, position 236 is closer to the external leaflet compared to the N-terminus of LptM near the lateral gate, indicating that LPS bypasses LptM during translocation.

Finally, having identified positions 236 and 239, respectively in β1 and β2, as sites efficiently crosslinked to LPS, we investigated whether

the N-terminal side of the LptD lateral gate slit serves as a preferential LPS-binding site. To test this, we introduced pBpa at positions A751 and G753 in β26 of LptD, at the C-terminal side of the lateral gate slit. These positions face T236 in β1 in the extended or contracted states of LptD, respectively. Crucially, whereas we could confirm efficient LPS crosslink for position T236, we did not detect LPS crosslinked to β26 (Supplementary Fig. 18). We conclude that the N-terminal side of the LptD lateral gate serves as a preferential LPS interaction site.

## Discussion

It has been unknown how the LptD β-barrel can open laterally to release LPS into the external leaflet of the OM. High resolution structures of LptD have shown extensive pairing between LptD β1 and β26, implying a high energetic barrier to lateral gate opening[24–26,34]. In our study, we provide evidence for an unprecedented conformation of the LptD β-barrel that can open via a mechanism facilitated by LptM. Our results suggest that the LptD lateral gate undergoes a conformational switch, which might explain how LPS reaches the membrane external leaflet.

We have previously identified LptM as a translocon component that is assembled with an LptD folding intermediate at the BAM complex. Predicted to bind near the β-taco/β-barrel interface, we showed that LptM promotes the subsequent step of LptD oxidative maturation characterized by the formation of inter-domain disulfide bonds. Given that LptM occupies a portion of the translocon near the predicted path of LPS transport, with this study we aimed at addressing how LptM influences translocon activity. Our biochemical and structural analyses identified a further translocon component, LptY, which binds the LptD β-taco domain externally to its LPS binding groove, as shown also by a recent study[43]. Differently from LptM, LptY does not appear to influence LptD oxidative maturation. We showed that, together, LptM and LptY stabilize the β-taco domain thereby facilitating the connection of the OM translocon to the IM components of the Lpt machinery (Fig. 6a). It remains to be determined whether the binding of LptY to LptD is regulated in vivo. Given that LptY presents a lipocalin-like fold and possesses two Cys residues predicted to form a disulfide bond, it is possible that a ligand or the periplasm oxidative state can influence the binding of LptY to LptD.

Remarkably, by stabilizing the LptD β-taco domain, LptY binding in the absence of LptM enabled the identification of a contracted LptD state, in which the β-taco domain and β1 of the β-barrel domain have converged, thereby contracting their interface. Indeed, in the contracted state β1 has slid along β26 of two amino acid positions toward the periplasm, revealing an unprecedented rearrangement of the lateral gate β-strands that provides direct evidence for their unzipping. Although we observed the contracted state in the absence of LptM, our functional assays with a LptD variant harboring crosslinkable Cys residues in β1 and β26 highlights the relevance of the contracted state for LptD function also in the presence of LptM. MD simulations of contracted LptD with modeled LptM showed enhanced lateral gate dynamics, which might explain why our cryo-EM structural analysis revealed the contracted LptD state only in the absence of LptM.

The LptM-induced lateral gate dynamics are likely to influence LPS transport. Our crosslinking results show that LptM remains associated with the translocon when LPS crosses the LptD lateral gate. The first 10 N-terminal residues of LptM interact with the internal wall of LptD β1-β4, whereas the C-terminal moiety of LptM was not structurally resolved, suggesting that it is highly dynamic. These results are in agreement with two recent structural studies[39,40] suggesting that LptM is a core component of the OM LPS translocon. We speculate that the C-terminal moiety of LptM can occupy the lumen of the β-barrel, as previously predicted by AlphaFold 2[36] but also move out of the translocon, e.g., by retrograde motion toward the periplasm. At the active translocon, displacement of the LptM C-terminal moiety would free the β-barrel lumen for access by LPS. Indeed, we identified luminal

oriented residues of LptD β1 and of the β1-β2 loop that can be cross-linked to LPS with strong efficiency, whereas β26 cannot be cross-linked with LPS, in agreement with a previous study[33]. An interaction at the apical region of β1 at the lateral gate slit is strongly affected by the lack of LptM, corroborating the role of this translocon component in functionalizing LptD. Further investigations are necessary to establish whether LptM is occasionally secreted by the LPS translocon as proposed by a recent study[40], nevertheless our data show that LPS can bypass LptM in the LptD β-barrel internal lumen.

Based on our structural and functional characterization of contracted LptD, we propose that the translocon operates by alternating between contracted and extended states. Contraction of LptD brings β1 of the β-barrel closer to the β-taco domain. At the lateral gate slit, β1 serves as a preferential LPS-interaction site, potentially facilitating LPS transfer from the β-taco to the β-barrel domain (Fig. 6b, Contracted-closed). In the contracted state, LptM enhances LptD lateral gate opening, thereby facilitating the insertion of the LPS acyl tails into the membrane and entry of its saccharides into the β-barrel lumen (Contracted-open). This step also requires that LPS crosses the luminal gate made of the β-taco/β-barrel hinge loop and a C-terminal segment of LptD[32]. We hypothesize that, at this stage, the acyl tails of LptM might help prevent the phospholipids of the OM inner leaflet from accessing the open lateral gate of LptD. By interacting with LPS and by repositioning during the contracted-to-extended state transition, β1 performs a stroke-like motion that can push LPS toward the external leaflet of the OM (Extended-open). Intriguingly, the connection of LptD to LptB$_2$FG via stacked β-taco domains suggests that ATP hydrolysis and LPS extraction at the IM could contribute to power the β-taco/β-barrel contraction/extension mechanism resulting in the stroke movement of β1 at the OM. With LptD in the extended-open state, repulsion of LPS phosphate groups by negatively charged residues in the translocon lumen[24,34] aids in the ejection of LPS through the lateral gate. The LPS release step can be further facilitated by LptE[48] and by interactions of LPS with divalent cations in the external leaflet. The translocon would then be ready to reset for the transport of a new molecule of LPS (Extended-closed). Additional studies are warranted to comprehensively demonstrate the proposed mechanism of LPS assembly.

The essential role of the Lpt machinery and its partial exposure to the cell surface make it an ideal target for the design of new antibiotics[49–51]. A detailed understanding of its components and their architecture to form a functional Lpt transenvelope machinery is of fundamental importance for drug-targeting. The Lpt components characterized in this study are both highly conserved in pathogenic species of *Enterobacteriaceae* and *Erwiniaceae*. In particular, LptM is also present in the threatening pathogens of the *Yersinia* and *Serratia* genera. Our results show that LptM and LptY enhance the efficiency of the Lpt pathway. However, it remains to be established whether their lack in other bacterial species is offset by compensatory mutations in alternative genes that might contribute to this pathway. In addition, and most strikingly, our results suggesting that LptD operates by alternating between contracted and extended states establish a detailed framework illustrating how LPS might be assembled at the surface of the OM. The improved understanding of the LPS translocon holo-complex has major implications for antibiotic development.

## Methods

### Bacterial strains and growth conditions

The *E. coli* strains used in this study are listed in Supplementary Table 3. All deletion strains were generated by transducing disrupted alleles into BW25113[52] using P1 phage lysates of the corresponding deletion strains in the Keio collection[53]. To obtain double deletion strains the Kanamycin-resistance (Kan$^R$) cassette introduced with the first transduction event was excised as previously described[36]. To delete chromosomal *lptD*, first BW25113 was transformed with a

**a**

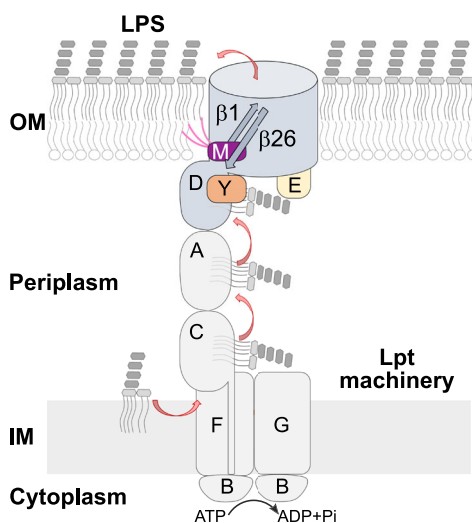

**b**

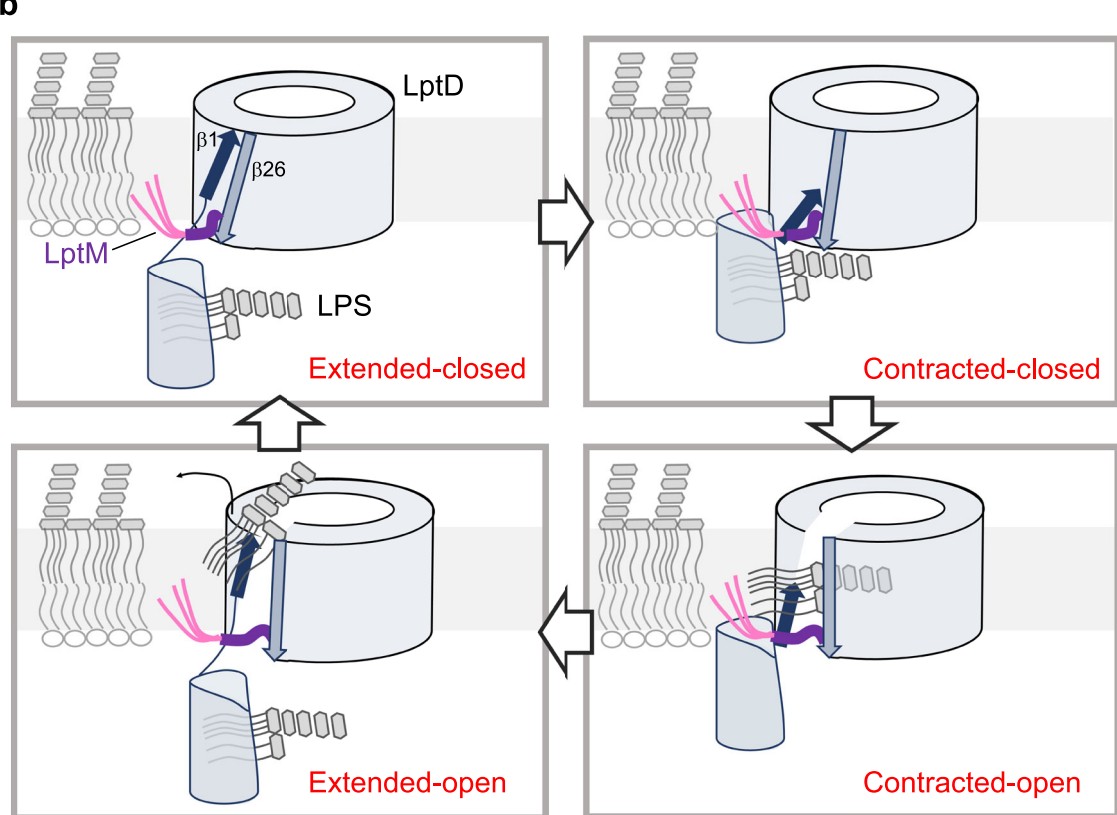

**Fig. 6 | LptD operates by switching between contracted and extended conformational states. a** Diagram of the Lpt pathway including the subunits LptM and LptY. **b** Mechanism of LPS insertion proposed on the basis of LptD conformational dynamics and functional assays described in this study. Top left LPS binds the β-taco domain of LptD (extended-closed state). Top right, LptD contracts, bringing LPS close to β1 of the LptD β-barrel domain (contracted-closed state). Bottom right, LptM facilitates lateral opening of LptD in the contracted state (contracted-open

state); the LPS sugars enter the lumen of LptD β-barrel, whereas its lipid tails move from the taco into the OM bilayer, bypassing LptM. Bottom left, LPS moves together with β1 towards the OM external leaflet, which restores the LptD extended state with an open lateral gate (extended-open state). LPS is then ejected from the LptD β-barrel domain, resetting LptD in the extended-closed state. For clarity, LptE and LptY are not shown.

plasmid expressing LptD (pLptDE^His or pLptDEM^His) thus generating a *lptD* diploid strain. Then, the chromosomal *lptD* gene was replaced with a Kan^R cassette using bacteriophage lambda recombination. A P1 phage lysate of the derived strain (lptD::Kan^R) was used to inactivate endogenous *lptD* by P1 transduction in *lptD* diploid strains harboring a plasmid-borne copy of wild-type *lptD* or mutated *lptD*. The obtained

transductants were reisolated on lysogenic broth (LB) agar plates prior to LptD functional tests measuring strain viability upon serial dilutions on LB agar containing or lacking the oxidant 4,4-dipyridyl disulfide (4-DPS).

*E. coli* strains were generally cultured on LB liquid media or LB agar plates without antibiotics or supplemented with one or more

antibiotics, including 100 µg/ml ampicillin, 50 µg/ml kanamycin, and 30 µg/ml chloramphenicol. Serial dilution assays were conducted on LB agar, LB agar supplemented with 0.05 mM 4-DPS, MacConkey agar supplemented with 120 µg/ml vancomycin, or 50 µM Isopropyl β-D-1-thiogalactopyranoside (IPTG), as indicated in the figures.

## Plasmid construction
Plasmids and oligonucleotides are listed respectively in Supplementary Tables 4 and 5. Plasmids harboring *lptY* alone (pLptY^His) or in operon with *lptD*, *lptE* or *lptM* (pLptDEY^His, pLptD^β-barrelEY^His) downstream of a P_lac derivative promoter were generated by integrating *lptY* in pTrc99a or in previously described vectors (pLptDEM^His, pLptD^β-barrelEM^His [36]) using a recombination-based, ligation-free cloning protocol. Plasmids encoding truncated forms of LptM were also generated by recombination-based cloning or by inverse PCR (see Supplementary Tables 4 and 5) using pLptDME^His as template. The replacement of specific *lptD* codons by an amber codon or by a cysteine-encoding codon was performed either by site-directed mutagenesis or by recombination-based cloning (see Supplementary Table 4).

## Site directed photocrosslinking
Site directed photocrosslinking was performed as previously described[36] using wild type and Δ*lptM* cells transformed with pEVOL-pBpF and variants of pLptDE^His or pLptDEM^His harboring amber codons at specific positions in the reading frame of *lptD* (Supplementary Table 4). Transformed cells were cultured until mid-exponential phase, supplemented with 1 mM pBpa and 200 µM IPTG for 1.5 h. Two identical cell culture aliquots were withdrawn, one was kept on ice protected from light and the other was subjected to UV irradiation (Tritan 365 MHB, Spectroline) for 10 min on ice. Envelope fractions prepared from both non-treated and UV irradiated cells were subjected to DDM-solubilization and nickel-affinity purification of LptE^His or LptM^His to isolate LptDE or LptDEM complexes as described above.

## LPS extraction and silver staining
*E. coli* strains containing empty vector (pCtrl) or pLptC were cultured in presence of 100 µM IPTG till OD600 = 0.5 to 0.7. Aliquots of cultures were harvested by centrifugation before freezing and LPS extraction by hot phenol-water method as previously described[36]. Aliquots of the LPS fraction was resolved by SDS-PAGE prior to silver staining using the SilverQuest Staining Kit (Invitrogen).

## Cell fractionation and isolation of protein complexes
Total cell lysates were prepared from cells cultured to mid-exponential phase at 37 °C. Where indicated, cells were supplemented with IPTG for 1.5 h prior to sample collection by centrifugation, lysis in Laemmli buffer and boiling. Protein affinity purification under mild-solubilization conditions was conducted as previously described[54]. Briefly, protein expression was induced in cell cultures at mid-exponential phase (OD600 = 0.5) by supplementing 200 µM IPTG for 1.5 h prior to cell collection by centrifugation. Cells were resuspended in 20 mM Tris-HCl pH 8 containing an EDTA-free protease inhibitor cocktail (Roche). This buffer was further supplemented with 50 mM iodoacetamide to alkylate free thiol groups in cysteine residues of proteins that had to be analysed for the presence of disulfide bonds. Resuspended cells were mechanically disrupted using a cell disruptor (Constant Systems LTD) set to 0.9 kPa. The obtained cell lysate was clarified by centrifugation at 6000 × *g*, 4 °C for 30 min. The crude envelope fraction was collected by subjecting the supernatant to ultracentrifugation at 50,000 x *g* at 4 °C for 30 min. The crude envelope fraction was solubilized with 50 mM Tris-HCl pH 8, 150 mM NaCl, 20 mM imidazole supplemented with EDTA-free protease inhibitor (Roche) and 1% (w/v) n-dodecyl-β-D-maltopyranoside (DDM, Merck). After a clarifying spin to remove insoluble material, solubilized proteins were incubated with Protino Ni-NTA resin (Machery-Nagel) for 2 h

at 4 °C. After extensive washes of the column with 50 mM Tris-HCl pH 8, 150 mM NaCl, 50 mM imidazole and 0.03% (w/v) DDM, bound proteins were eluted with a similar buffer containing 800 mM imidazole. Aliquots of the elution fractions supplemented with 10% (w/v) glycerol were snap-frozen in liquid nitrogen for storage at −80 °C or directly analyzed by SDS or blue native gel electrophoresis. The purification of LptDEM^His (yielding also LptDEM^HisY) or LptDE^His (yielding also LptDE^HisY) for cryo-EM was performed from 2-liter cultures of a wild-type or a Δ*lptM* strain carrying respectively pLptDEM^His or pLptDE^His. Membranes were prepared and solubilized in DDM as describe above. Protein complexes were first purified by nickel-affinity chromatography and then subjected to size exclusion chromatography (Superdex 200 increase 10/300, Cytiva) in 50 mM Tris-HCl pH 8, 50 mM NaCl and 0.03% (w/v) DDM. Fractions of interest were collected and concentrated to 0.3 mg/mL of proteins by ultrafiltration with a molecular weight cut-off of 100 kDa (Millipore). Fresh protein samples were immediately used to prepare cryo-EM grids.

## Gel electrophoresis and protein detection
Proteins samples were diluted in Laemmli buffer lacking β-mercaptoethanol (non-reducing conditions) or supplemented with β-mercaptoethanol (reducing conditions). Proteins were separated by home-made SDS polyacrylamide gels (10% acrylamide in Bis-Tris pH 6.4 buffer, subjected to electrophoresis using either MES or MOPS buffer). Where indicated, gels were stained with Coomassie Brilliant Blue R250. To perform Western blots, protein gels were blotted onto PVDF membranes and subjected to immunodetection using epitope-specific rabbit polyclonal antisera or with an anti-LPS mouse monoclonal antibody (WN1 222-5, Hycult Biotech). Immunodetection was revealed by using enhanced chemiluminescence. The signal intensity of protein bands was quantified using Image Lab software (BioRad). The rabbit polyclonal antiserum against LptD and against LptA were kind gifts of Drs. J.F. Collet (Louvain, Belgium) and A. Polissi (Milan, Italy), respectively.

## Native MS
Mass spectra presented in Supplementary Fig. 4a, c were obtained by re-analysing data previously deposited to the ProteomeXchange Consortium via the PRIDE partner repository[55] with the dataset identifier PXD041774. Mass spectra presented in Supplementary Fig. 4b, d were obtained by analysing data deposited to the ProteomeXchange Consortium via the PRIDE partner repository with the dataset identifier PXD068376. Mass spectra presented in Supplementary Fig. 4b, d and showing the dissociated monomers were acquired with similar parameters as in ref. [36], but with an extended m/z range of 1500–3000 Th. Raw data were acquired with MassLynx 4.1 (Waters, Manchester, UK) and analysed manually.

## Cryo-EM sample preparation and data acquisition
3 µL of sample were deposited onto glow-discharged R2/1 carbon grids with an additional 2 nm ultrathin continuous carbon layer and placed in the thermostatic chamber of a Leica EM-GP automatic plunge freezer, set at 4 °C and 95% humidity. Excess solution was removed by blotting with Whatman n°1 filter paper and the grids were immediately flash frozen in liquid ethane at −185 °C. Images were pre-screened in house on a Talos Arctica (Thermo Fisher Scientific) operated at 200 kV in parallel beam condition with a K3 Summit direct electron detector and a BioQuantum energy filter (Gatan Inc.). Energy-filtered (20 eV slit width) image series were acquired with Digital Micrograph software.

Data collection was performed at EMBL Heidelberg on a Titan Krios G4 microscope equipped with a cold field emission gun (C-FEG) operated at 300 kV. Images were recorded using a Falcon 4i direct electron detector in counting mode, coupled to a SelectrisX energy filter with a 10 eV energy slit width. SerialEM (version 4.1.0beta)[56] was used for automated data collection at a nominal magnification of

165,000x in nanoprobe mode (spot size 5). The microscope was operated with a 50 μm C2 aperture and a 100 μm objective aperture, with a beam diameter of 480 nm and a spherical aberration (Cs) of 2.7 mm. This resulted in a calibrated pixel size of 0.731 Å at the specimen level.

## Image processing

All image processing steps were performed in cryoSPARC v4[57]. EER movies from both datasets were imported and processed using Patch motion correction, while CTF parameters were estimated using Patch CTF estimation. After initial inspection, micrographs showing significant drift, poor CTF fits, or ice contamination were discarded.

## Determination of LptDE and LptDEY structures

Initial data preprocessing was performed on 20,832 micrographs in cryoSPARC. After CTF estimation and manual curation, 20,789 micrographs were selected for further processing. Initial particle picking was performed using a blob picker with a diameter range of 100–150 Å, yielding 7,573,562 particles, from which 5,280,207 particles were retained after inspection. A subset of 1,116,177 particles from 5000 micrographs was extracted with a box size of 400 pixels for initial processing.

After 2D classification into 100 classes, 337,670 particles (30%) were selected and further classified into 50 classes, yielding 180,417 particles (53%). Based on these initial classes, a second round of template-based particle picking was performed using a 170 Å diameter, resulting in 11,445,246 particles. After inspection and cleanup, 5,219,710 particles were extracted. These particles underwent multiple rounds of 2D classification: first into 100 classes yielding 1,916,647 particles (37%), then into 50 classes resulting in 829,104 particles (43%), and finally to 229,126 particles (28%) showing high-quality features.

The selected particles were subjected to ab initio reconstruction followed by homogeneous refinement, reaching 4.25 Å resolution. After local refinement, the resolution improved to 2.89 Å. The particles were then re-extracted with a 360-pixel box size and underwent multiple rounds of heterogeneous refinement, progressively improving the particle set through 187,497, 144,168, and finally 86,702 particles.

A final round of template-based particle picking was performed with a 160 Å diameter. After extraction and extensive 2D classification, a set of 457,205 particles was obtained and refined to 3.18 Å, with local refinement improving the resolution to 2.44 Å. Ab initio reconstruction into four classes followed by further refinement produced the final LptDE reconstruction at 2.74 Å resolution from 107,410 particles. Resolution was estimated using the FSC = 0.143 criterion.

A parallel processing branch starting with 829,104 particles underwent heterogeneous refinement yielding 361,218 particles, then 254,562 particles, and finally 187,497 particles. After additional refinement and 2D classification steps, 86,702 particles were obtained. A final round of template picking was performed using a 160 Å diameter. The resulting 13,173,675 particles underwent multiple rounds of processing, including extraction and binning (360 px → 180 px) of 6,375,528 particles, followed by successive 2D classifications with 200 and 100 classes. The final set of 127,785 particles was refined to 2.94 Å, with local refinement yielding the final LptDEY reconstruction at 2.62 Å resolution. Resolution was estimated using the FSC = 0.143 criterion.

## Determination of LptDEM and LptDEMY structures

Initial data preprocessing of 18,816 micrographs was performed in cryoSPARC. After manual curation, 18,287 micrographs (97%) were selected for further processing. Template-based particle picking was performed using a diameter of 250 Å, yielding 2,406,295 particles after inspection and extraction with a box size of 360 pixels. The extracted particles underwent two rounds of 2D classification. The first round with 100 classes yielded 909,602 particles (38%), which were further

classified into 50 classes, resulting in a selection of 510,229 particles (56%). These particles were subjected to heterogeneous refinement with two volumes, and one population of 273,204 particles was refined to 2.86 Å using non-uniform refinement. After further classification and refinement steps, a set of 302,650 particles underwent three rounds of heterogeneous refinement, progressively improving the quality of the reconstruction. The final population of 198,109 particles was refined using homogeneous refinement to 2.63 Å, followed by local refinement that yielded the final LptDEM reconstruction at 2.47 Å resolution. Resolution was estimated using the FSC = 0.143 criterion. For LptDEMY, further optimization focused on the dataset containing 302,650 particles. It underwent 3D classification into two classes using a mask focusing on the known location of LptY in the LptDEY structure. The particles belonging to the class displaying a density at the location of LptY (146,385 particles) were selected for final non-uniform refinement, yielding a reconstruction at 2.63 Å resolution. Resolution was estimated using the FSC = 0.143 criterion.

## Model building and refinement

Initial models for LptD and LptE were generated using AlphaFold2 predictions[41] and rigid-body fitted into the experimental density maps. These initial models were modified through multiple rounds of manual rebuilding in ISOLDE[58] and COOT[59] to accurately fit the experimental density. For model building, maps were sharpened using DeepEMhancer[60] to enhance interpretability of structural features. LptM was built de novo into the corresponding density. For LptY-containing complexes, an AlphaFold2-predicted model was rigid-body docked into the experimental density (correlation score in ChimeraX of 0.439 for LptDEY and 0.473 for LptDEMY).

For real-space refinement in PHENIX[61], maps were sharpened using phenix.autosharpen. The models underwent multiple rounds of manual refinement in ISOLDE and real-space refinement using phenix.real_space_refine with appropriate geometric restraints. Final models were validated using MolProbity[62] and phenix.validation_cryoem implemented in the PHENIX software. Details about the cryo-EM refinement statistics and FSC Map versus Model plot can be found in Supplementary Table 6 and Supplementary Figs. 6 and 7.

## Focused processing of LptY density

To improve the resolution of the LptY binding region, we attempted several focused processing approaches in CryoSPARC. These included: (1) focused refinement with a soft mask encompassing the LptY-LptD β-taco interface region, (2) 3D classification without alignment focused on the LptY density to identify conformational heterogeneity, and (3) 3DFlex reconstruction to account for potential flexibility in the LptY binding region. Despite these efforts, none of these approaches yielded improved density quality for LptY, likely reflecting the inherent flexibility of this lipoprotein component when bound to the translocon complex.

## Molecular modeling and dynamics simulations

LptM was added to an extended complex of LptDEY through superimposition of the extended and contracted states. The aligned coordinates of LptM were taken from the extended conformation and added to that of the contracted state. The final complex was then subjected to energy minimization using GROMACS[47]. Complexes of LptDE, LptDEM, LptDEY, and LptDEMY in both contracted and extended states were assembled into model outer membranes using our MemProtMD protocol[63]. In brief, the systems were converted to a Coase-Grained (CG) Martini 3 force field representation using Martinize2[64,65]. Here an elastic network with a force constant of 1000 kJ mol⁻¹ nm⁻² was used to connect Cα atoms within 10 Å in the protein structure. The protein was placed into a model outer membrane using *insane*, with 100% Kdo2-lipid A (KLA) in the outer leaflet and a 70:20:10 ratio of 1-palmitoyl-2-oleoyl-sn-glycero-3-

phosphoethanolamine (POPE), 1-palmitoyl-2-oleoyl-sn-glycero-3-phosphoglycerol (POPG) and cardiolipin in the inner leaflet. The system was then solvated and ionized with 0.15 M NaCl. The system was equilibrated in CG before conversion to a CHARMM36m[66] representation using CG2AT[67]. Three repeats of 500 ns atomistic MD simulations were performed with a timestep of 2 fs for each LptDE assembly and state. All simulations were performed in the isothermal-isobaric ensemble at 310 K and 1 bar. Pressure was maintained at 1 bar with an semi-isotropic compressibility $3 \times 10^{-4}$ using the C-rescale barostat[68]. Temperature was controlled using the velocity-rescale thermostat[69], with the solvent, lipids and protein coupled to an external bath. All MD simulations were performed using GROMACS 2023[47] with Root Mean Standard Fluctuations (RMSF), interface Solvent Accessible Surface Areas (SASA), membrane thickness, Hydrogen-bond calculations, gate distances, and dihedral angles all analysed using GROMACS tools and MDAnalysis[70].

All images were generated using PyMOL[71]. MD simulations data are available for all Lpt states and complexes for start (0 ns) and end (500 ns) frames for each simulation repeat (Zenodo repository, [https://doi.org/10.5281/zenodo.16643358]). Starting tpr files are also available, and xtc trajectories can be supplied on request. Information regarding reproducibility of the MD simulations is reported in Supplementary Table 7.

### Additional Methods
Additional methods used in this study are described in the Supplementary Information.

### Reporting summary
Further information on research design is available in the Nature Portfolio Reporting Summary linked to this article.

## Data availability
LptDE model information is available under accession PDB 9I9Z and the density map under accession EMDB 52773. LptDEY map is available under EMDB 52777 and the pertaining LptDE model under accession PDB 9IA0. LptDEM model information is available under accession PDB 9IA2 and the density map under accession EMDB 52778. LptDEMY map is available under EMDB 52779 and the pertaining LptDEM model under accession PDB 9IA5. MD simulations source data are available at the Zenodo repository, [https://doi.org/10.5281/zenodo.16643358]. Source data relative to the native MS analysis have been deposited to the Pride partner repository with the dataset identifier PXD041774 and PXD068376. LC-MS/MS source data have been deposited to the Pride partner repository with the dataset identifier PXD068376. Other source data are provided as a Source Data file. These include source data used for the phylogenetic analysis, source data relative to the LC-MS/MS analysis in Supplementary Fig. 1b and to the densitometry quantifications of Fig. 5e, uncropped scans of all blots and gels in Figures. Uncropped scans of all blots and gels in Supplementary Figs. are supplied in Supplementary Fig. 19. Source data are provided with this paper.

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

## Acknowledgements

Research in R.I.'s laboratory was funded by the Agence Nationale de la Recherche (ANR-23-CE11-0025-01), the China Scholarship Council fellowships to H.C., the FRM PhD fellowships to Y.A.A. (FRM ECO202106013640) and the FRM Fourth-Year PhD fellowship to H.C. (FRM FDT202404018564). Cryo-EM imaging of samples was

sponsored by the iNEXT Discovery program (PID: 22756) to R.I. Research in R.F.'s laboratory was funded by the Agence Nationale de la Recherche (ANR-21-CE44-0002) and the CNRS. We acknowledge the METi imaging facility, member of the national infrastructure France-BioImaging supported by the Agence Nationale de la Recherche (ANR-10-INBS-04). Research in J.M. laboratory was supported by the Agence Nationale de la Recherche: ANR-23-CE11-0025-01 and ProFI projects (ANR-10-INBS-08 & ANR-24-INBS-0015). P.J.S. acknowledges the NIH (R01AI174416 (PI: M. Stephen Trent)), Wellcome (208361/Z/17/Z), MRC, BBSRC, EPSRC and the Howard Dalton Centre for funding. P.J.S. and D.G. acknowledge Sulis at HPC Midlands + , which was funded by the EPSRC on grant EP/T022108/1, and the University of Warwick Scientific Computing Research Technology Platform for computational access. This project made use of time on ARCHER2 granted via the UK High-End Computing Consortium for Biomolecular Simulation, HECBioSim (http://www.hecbiosim.ac.uk), supported by EPSRC (grant no. EP/R029407/1).

## Author contributions

H.C. and V.M. performed genetic analyses, protein purifications, and functional assays, with contributions from Y.A.A. and A.C.-S.; S.B. prepared cryo-EM grids; R.F. and A.S. determined the cryo-EM structures; P.J.S. performed structural modeling and MD simulations, and analyzed the data together with D.G.; Y.Q. performed phylogenetic analysis; J.M. performed native MS; J.M., M.M., and C.F. performed LC-MS/MS analysis; H.C., A.S., V.M., Y.Q., J.M., and D.G. prepared the figures; R.I., R.F., and P.J.S. wrote the manuscript with contributions from J.M., Y.Q., V.M., and A.C.-S.; R.I. conceived the research. R.I., R.F., and P.J.S. supervised the work and acquired funding. All authors discussed the results and commented on the manuscript.

## Competing interests

The authors declare no competing interests.
