## [Transparent Peer Review file · Nature Communications]

Structural Basis of Lipopolysaccharide Assembly by the Outer Membrane Translocon Holo-Complex

Corresponding Author: Dr Raffaele Ieva

Version 0:

Reviewer comments:

Reviewer #1

(Remarks to the Author)

In the manuscript from Chen et al, the authors provide new structures and investigations of the outer-membrane part of the Lpt machinery. In particular, they describe the structures and functions of two associated proteins, LptM and LptY. They find that LptY stabilizes the periplasmic domain of LptD while LptM inserts into the lateral gate of LptD, enhancing opening. These results provide further insight into how LPS are inserted into the OM.

This is a very interesting and enlightening study regarding Lpt transport. I have a few requests for further analysis of the MD simulations.

Regarding the MD simulations, I understand the focus on LptD and the gate in particular, but I think it would be helpful to see more on the stability of LptM and LptY as well as their interactions with LptD. Do they move at all? Are the number of hydrogen bonds, etc. the same throughout the runs? An SI figure might be appropriate here.

Additionally, I am glad the authors ran three replicas of each simulation, but except for Fig. S14 (and the really nice movie!), I didn't notice any data on these presented. For example, could these be used to provide standard deviations for the distances in Figure S13 (I assume the average is over all three replicas)? Similarly, hydrogen bond counts for the gate presented in the main text could benefit from this. Any other places the authors see fit to highlight differences (or similarities) between the runs would be appreciated.

Do the authors observe any membrane thinning or destabilization near the LptD gate in simulations that would aid LPS insertion? If so, how does it depend on the presence of LptM, in particular? A little more analysis and discussion of this would be informative.

Figure S13: The end of the caption says "in in a)".

Reviewer #2

(Remarks to the Author)

The LptDE complex plays a central role in delivering lipopolysaccharide (LPS) to the bacterial outer membrane (OM), a process that has attracted significant research attention. Despite progress, the conformational changes underlying LPS insertion and the possible involvement of auxiliary proteins remain poorly defined.

In this study, Chen et al. build on their previous work to elucidate the mechanism of LptDEM-mediated LPS transport and identify a previously uncharacterized lipoprotein, LptY (formerly YedD), which interacts with LptDE to form a stable LptDEMY holo-complex. Using cryo-electron microscopy (cryo-EM), the authors resolve structures of LptDE in complex with LptM, LptY, and the full trimeric holo-complex. Structural analyses reveal that LptM and LptY both stabilize the β -taco domain of LptD, with LptY binding the periplasmic β -taco domain to reinforce its function as an LPS receptor, while LptM intercalates into the lateral gate of the β -barrel, promoting its opening for LPS entry.

Crucially, the study identifies two distinct conformational states at the β -taco/ β -barrel interface—contracted and extended—marking the first direct observation of such transitions. β -strand 1 of LptD, a key element of the lateral gate, undergoes a "stroke-like" motion toward the outer membrane's external leaflet during LPS binding, driving the transition from the

contracted to the extended state.

Together, these findings offer a detailed mechanistic model for the selective transport of LPS to the OM and establish a theoretical framework for understanding the asymmetric assembly of the outer membrane. The study's integration of cryo-EM, functional assays, and molecular dynamics simulations represents a significant advancement in the field.

Minor points:

1. The cryo-EM density map for the LptY binding site appears to be of relatively low resolution, which may compromise the accuracy of the modeled interactions in this region. The authors are encouraged to provide a locally refined density map or apply focused classification to improve the resolution and enhance confidence in the structural interpretation.
2. In Figure 3, the deletion of LptM and LptY under LptC overexpression conditions results in impaired bacterial growth on MacConkey agar. The authors should clarify whether this phenotype can be directly attributed to a reduction in LPS transport efficiency, as this would strengthen the functional relevance of the LptM/Y components.
3. The manuscript notes that LptM and LptY are present only in a subset of bacterial strains. This raises the question of functional redundancy in the Lpt pathway. The authors should discuss whether the absence of these components in some strains implies compensatory mechanisms and whether gene knockouts lead to LPS accumulation in the inner membrane, indicating disrupted transport.
4. The bacterial growth experiment in Fig 4e used the streak plate method, while the other related experiments employed the dilution method. It is recommended that the authors standardize the method. On the other hand, the formation of disulfide bonds after mutation was not assessed in Fig 4e, making it impossible to determine whether the protein correctly forms disulfide bonds after the mutation.
5. The final sections of the Results contain repetitive descriptions of LptM function. To improve clarity and flow, the authors are advised to streamline these sections and consolidate overlapping findings.

Reviewer #3

(Remarks to the Author)

The biogenesis of the outer membrane of Gram-negative bacteria requires the transport of bulky lipopolysaccharides (LPS) from the periplasmic face of the inner membrane to the outer leaflet of the outer membrane, a process that is accomplished by the critically important Lpt pathway. Chen and co-workers describe the structural and functional characteristics of LptM and LptY (YedD), two recently discovered accessory factors to the LptD-LptE outer membrane core complex of the LPS translocon. Whilst dispensable under optimal growth conditions, both accessory factors are shown to be required under conditions where Lpt pathway activity becomes limiting.

The authors report the cryoEM structure of *E. coli* LptDE, LptDEY, LptDEMY and LptDEM complexes, and identify two distinct conformational states in LptD, representing an 'extended' and 'contracted' conformation of the periplasmic b-taco domain and its contact with the lateral gate of the b-barrel domain. These observations are followed up with Cys-Cys and pBpA cross-linking mutants to evaluate the physiological relevance of the conformational states, and their implications for the LPS transport mechanism.

Based on these experiments, the authors propose a transport model that invokes conformational cycling between the extended and contracted states to facilitate the passing of LPS molecules from the b-taco to the LptD lumen, and then onwards to the OM outer leaflet.

LPS molecules reach the b-taco domain of the LptDE translocon via the periplasmic LptA bridge and the activity of the inner membrane complex LptB2FG. The mechanism by which the bulky LPS molecules are then translocated to the cell surface and inserted into the OM outer leaflet remains unclear. The discovery and characterization of the LptM and LptY accessory factors will undoubtedly form an important contribution to the functional studies of the LptDE translocon.

This new study is generally well documented and provides solid data to the structural and functional characteristics of LptM and LptY complexes of the LptDE translocon. Although the translocation model proposed by the authors is plausible and in agreement with their experimental data, it remains speculative and would require additional data and/or a more balanced discussion. I'm supportive of publication of this new study provided that some points are clarified by the author.

The main point to address is whether the contracted state of the LptDE taco domain – b1-b26 gate is physiologically relevant, or an artefact of a LptDE complex lacking LptM.

Specific comments:

- By comparison of LptDE and LptDEM and LptDEMY complexes, the authors identify a yet undocumented 'contracted' state in the LptDE complex, which in their study correlates with the lack of LptM. From the authors data (e.g. Ext. Data Fig. 9d), it would appear that LptM is a constitutive component of the LptDE translocon. When comparing published LptDE structure from *Klebsiella* and *Shigella*, and the *E. coli* LptDE-YedD complex, the authors show that these fit the extended, LptM-bound state. Did the authors revisit the deposited density maps to check for the presence of unassigned density that may allude to the presence of LptM? Whilst the authors cannot experimentally validate the presence of LptM in these complexes, it would be important to revisit the reported structure and comment on the likelihood they represent LptM bound complexes. i.e. should the field consider LptM a core component of the LptDE complex? It would appear so.

- Further to previous point, the authors should seek further evidence that the contracted state is physiologically relevant and not an artefact of a LptDE complex lacking LptM. They show that 4-DPS induced oxidation of LptD A233C/I756C and K234C/N755C mutants, which are juxtaposed Cys pairs in the contracted conformation, results in a loss of function. However, can the authors conclude beyond doubt that the loss of function is due to trapping the LptDE(M?) complex in the contracted state versus resulting in a folding defect on the BAM complex? The authors should do a pulldown of the LptD A233C/I756C and K234C/N755C mutants to confirm that these indeed represent fully folded LptDE in contracted state – similar to what is done in Ext. Data Fig. 9d. Moreover, the authors should test for the presence of LptM in case of contraction - locked mutants A233C/I756C and K234C/N755C to address whether the contracted state is compatible with the binding of LptM.

- In extended data Fig. 12 the authors show the RMSF of the various LptDE complexes with and without LptM and LptY, both in the extended and contracted states. None of the in silico generated states (LptDE extended conformation, or contracted conformation of LptM containing complexes) relaxes to the state seen experimentally, or shows an increased RMSF in the taco domain and b1-b26 gate. The authors should comment on this point.

- The authors see an LptM-dependent difference in LPS cross-linking at T236, which they use in support of their argument that LptM increases conformational dynamics of the b-taco – b1-b26 gate contact. How do the authors interpret the lack of LptM responsiveness of the N239 position?

- The authors should take a more reserved position in abstract and discussion in the presentation of the proposed mechanistic model. Whilst plausible, an experimental validation of the model is lacking. In their discussion, the authors do not comment on the envisaged function of LptY.

Minor points

- Fig. 1. What density is shown, and at what contour level? There is no mention of this in the legend. What is the purpose of the density shown? It does not fit the complexes, or the density shown in Ext Data Fig. 6, 8 or Figure 2, but rather looks like carved out micelles. As is, this is misleading and of little use, or should be explained in Figure / main text.
- Ext. Data Fig. 1a. Presumably the LptDEHis sample results from pLptDE-his expression in a LptM knockout-strain? Clarify this in the legend. The lack of LptM now looks contradictory to Ext. Data Fig. 9d, where expression of pLptDE-his in a WT vs lptM knockout background does result in a strong LptM signal in the IMAC pulldown.
- Main text p6 and Ext. Data Fig. 5 describe the co-elution of LptD and LptM with LptY pull-downs. Why is LptE not included in the Western or Coomassie analysis? Presumably the pulled complexes include LptE?
- Ext. Data Fig. 5. On what basis do the authors assign the band indicated with * as a non-specific reaction? How come this band indicated as non-specific is lacking in the Ext. Data Fig. 8a?
- Ext. Data Fig. 5. It would be helpful to add an anti-His stain for the 15 kDa range.
- In p8-9 and Fig. 3 and Ext. Data Fig. 9a, the authors comment on the fitting of the LptY AlphaFold model in the cryoEM map despite their low resolution. Please provide the CC values for the model – map fits.
- Ext. Data Fig. 9d. How do the authors interpret the pulldown of the BamD in absence of LptM, though without the other BAM subunits co-eluting?
- Methods describe the purification of LptDEM and LptDE, but not of LptDEMY or LptDEY. This should be added.
- Extended Data Fig. 13. Better clarify how the distances after measured. What atoms are used to calculate the distances and in a residue pair x and y, what is the difference between the x-y and y-x distance?

Version 1:

Reviewer comments:

Reviewer #1

(Remarks to the Author)

The authors have done a thorough job addressing all reviewer comments.

Reviewer #2

(Remarks to the Author)

Reviewer #3

(Remarks to the Author)

In response to the reviewers' comments, the authors add an extensive set of new results and additional figure panels. These further strengthen this excellent study and take away most of the uncertainty that was associated with some of the functional models proposed.

A key point to address was the assessment of the physiological relevance of the contracted state of the LptDE translocon, i.e. 1) determining its existence in presence of LptM and 2) determining whether it is an on pathway conformation, or might be an off-pathway artefact from the lptM knockout in which it was identified. The revised manuscript adds experiments to

address these points.

Most convincingly, the authors show that the 4-DPS-dependent loss of function of LptD N232C/E757C, A233C/I756C or K234C/N755C, which cross-links b1-b26 in the contracted state, is independent of the presence or absence of LptM, suggesting that the contracted state is also sampled in complexes containing LptM. Furthermore, 4-DPS lethality is dependent on Cys pairs representing the contracted state. Single Cys mutants proof insensitive to 4-DPS. The authors further demonstrate that b1-b26 mutants N232C/E757C, A233C/I756C or K234C/N755C do not accumulate on the BAM complex and are compatible with LptM binding. However, in both cases this was done under reducing conditions, which does not inform on the conformational state of the mutants.

The authors argue these same experiments cannot be done under oxidizing conditions (4-DPS) because of the toxic effect of locking the b1-b26 gate. Can this not be circumvented by adding 4-DPS at higher cell density, so that sufficient cell mass can be harvested for pull-downs and Western analysis? This smaller experiment would still substantially strengthen the authors' proposed model that sampling the LptDE contracted state occurs in fully functional LptDEMY complexes and is part of the translocation activity.

We thank the Reviewers for the favourable evaluation of our study and for recognizing the new advances it offers in understanding the mechanism of LPS assembly into the OM. Following their constructive comments, we have expanded some of our analyses and performed several new experiments resulting in 3 modified figure panels (Suppl. Fig. 2 with improved labelling, Suppl. Fig. 5 with additional blots, Suppl. Fig. 15b with expanded statistical analyses), and 17 new figure panels (Fig. 3e; Fig. 4f; Suppl. Fig. 8d; Suppl. Fig. 12a-c; Suppl. Fig. 14 a-j; Suppl. Fig. 17), 32 new figure panels relative to the MD simulation analyses that have been deposited on the Zenodo repository (<https://doi.org/10.5281/zenodo.16643358>) and the Figure panels A-J associated to our point-by-point response. Finally, we have included our uncropped gels and drop test images in Suppl. Figs 18 and 19 of the revised manuscript.

We are grateful to the Reviewers also for highlighting different points that needed further clarification.

Please find below our point-by-point response to the Reviewers' comments.

Reviewer #1 (Remarks to the Author):

In the manuscript from Chen et al, the authors provide new structures and investigations of the outer-membrane part of the Lpt machinery. In particular, they describe the structures and functions of two associated proteins, LptM and LptY. They find that LptY stabilizes the periplasmic domain of LptD while LptM inserts into the lateral gate of LptD, enhancing opening. These results provide further insight into how LPS are inserted into the OM.

This is a very interesting and enlightening study regarding Lpt transport. I have a few requests for further analysis of the MD simulations.

Regarding the MD simulations, I understand the focus on LptD and the gate in particular, but I think it would be helpful to see more on the stability of LptM and LptY as well as their interactions with LptD. Do they move at all? Are the number of hydrogen bonds, etc. the same throughout the runs? An SI figure might be appropriate here.

1- From all of the simulations, we have assessed per subunit residue fluctuations (RMSF) and secondary structure retention. We have included both means of the replicas and the standard deviation from the 3 repeats (these data are provided as figure files in the Zenodo repository including the MD simulation data, <https://doi.org/10.5281/zenodo.16643358>). We have also assessed the interface surface area between subunits over the course of the three simulation repeats that show that while there are dynamics at the interface, the surface area remains similar over the course of the simulation (new Suppl. Fig. 14a-j). Note that, as these proteins are membrane embedded, there are fewer hydrogen bonds to track at the interfaces.

Additionally, I am glad the authors ran three replicas of each simulation, but except for Fig. S14 (and the really nice movie!), I didn't notice any data on these presented. For example, could these be used to provide standard deviations for the distances in Figure S13 (I assume the average is over all three replicas)? Similarly, hydrogen bond counts for the gate presented in the main text could benefit from this. Any other places the authors see fit to highlight differences (or similarities) between the runs would be appreciated.

2- We have added the standard deviation values to Suppl. Fig. 15b of the revised manuscript (Suppl. Figure 13 of the original submission). We have also transferred the standard deviation

values from Suppl. Figure 15a (Suppl. Figure 13a of the original submission) to the main body as suggested.

Do the authors observe any membrane thinning or destabilization near the LptD gate in simulations that would aid LPS insertion? If so, how does it depend on the presence of LptM, in particular? A little more analysis and discussion of this would be informative.

3- We have looked into this, as it is very much of interest. While there are ripples in the membrane that thin or thicken the lipid bilayer around the β -barrel of LptD, there are no obvious differences at the gate (see Figure attached at the end of our response letter). Even if it appears sensible that the membrane would be disrupted as LPS is inserted, we do not see this effect during the time course of our MD simulations.

Figure S13: The end of the caption says "in in a)".

4- This has been corrected.

Reviewer #2 (Remarks to the Author):

The LptDE complex plays a central role in delivering lipopolysaccharide (LPS) to the bacterial outer membrane (OM), a process that has attracted significant research attention. Despite progress, the conformational changes underlying LPS insertion and the possible involvement of auxiliary proteins remain poorly defined.

In this study, Chen et al. build on their previous work to elucidate the mechanism of LptDEM-mediated LPS transport and identify a previously uncharacterized lipoprotein, LptY (formerly YedD), which interacts with LptDE to form a stable LptDEMY holo-complex. Using cryo-electron microscopy (cryo-EM), the authors resolve structures of LptDE in complex with LptM, LptY, and the full trimeric holo-complex. Structural analyses reveal that LptM and LptY both stabilize the β -taco domain of LptD, with LptY binding the periplasmic β -taco domain to reinforce its function as an LPS receptor, while LptM intercalates into the lateral gate of the β -barrel, promoting its opening for LPS entry.

Crucially, the study identifies two distinct conformational states at the β -taco/ β -barrel interface—contracted and extended—marking the first direct observation of such transitions. β -strand 1 of LptD, a key element of the lateral gate, undergoes a "stroke-like" motion toward the outer membrane's external leaflet during LPS binding, driving the transition from the contracted to the extended state.

Together, these findings offer a detailed mechanistic model for the selective transport of LPS to the OM and establish a theoretical framework for understanding the asymmetric assembly of the outer membrane. The study's integration of cryo-EM, functional assays, and molecular dynamics simulations represents a significant advancement in the field.

Minor points:

1. The cryo-EM density map for the LptY binding site appears to be of relatively low resolution, which may compromise the accuracy of the modeled interactions in this region. The authors are encouraged to provide a locally refined density map or apply focused

classification to improve the resolution and enhance confidence in the structural interpretation.

1- We appreciate the Reviewer's concern regarding the resolution of the LptY binding region. We acknowledge that the LptY density shows lower local resolution compared to the transmembrane components. We attempted multiple computational approaches to improve the LptY density quality, including focused refinement, 3D classification, and flexibility correction (3DFlex in CryoSPARC). Unfortunately, the densities for LptY could not be improved by any of these approaches (see our modification to the Methods of the revised manuscript, paragraph entitled "Focused processing of LptY density"). The limited resolution of LptY likely reflects the inherent flexibility of this lipoprotein in its translocon-bound state.

2. In Figure 3, the deletion of *lptM* and *lptY* under *LptC* overexpression conditions results in impaired bacterial growth on MacConkey agar. The authors should clarify whether this phenotype can be directly attributed to a reduction in LPS transport efficiency, as this would strengthen the functional relevance of the *LptM/Y* components.

2- Following the suggestion of the Reviewer, we have extracted LPS from the wild-type, Δ lptM, Δ lptY, and Δ lptM Δ lptY strains, upon interfering with the Lpt bridges by overproducing LptC. SDS-PAGE and silver staining shows that LPS extracted from the double deletion strain (sensitive to McConkey agar) presents slower migrating forms. Typically, slower-migrating forms of LPS are obtained when LPS transport to the OM is strongly impaired, leading to accumulation of LPS molecules at the inner membrane, where these can be modified by the O-antigen ligase^{1,2}. This new result (illustrated in Fig. 3e of the revised manuscript) further supports our model according to which LptM and LptY together promote optimal functionality of the Lpt transenvelope bridges.

3. The manuscript notes that *LptM* and *LptY* are present only in a subset of bacterial strains. This raises the question of functional redundancy in the Lpt pathway. The authors should discuss whether the absence of these components in some strains implies compensatory mechanisms and whether gene knockouts lead to LPS accumulation in the inner membrane, indicating disrupted transport.

*3- The results of our phylogenetic analyses are consistent with a relatively recent appearance of the genes encoding LptM and LptY. These genes can be associated with a gain of function, such as regulation of LPS transport, as suggested by the result of Fig. 3d and e of our revised manuscript. The absence of these genes in other genomes may suggest that regulation of LPS transport is less critical for the fitness of these species. However, it cannot be ruled out that non-homologous genes in other genomes perform functions similar to those of LptM and LptY. In addition, our LptY protein tree illustrated in Suppl. Fig. 2 shows that, once these genes have been acquired, they can be subsequently lost. Indeed, in the genomes of some bacteria engaged in symbiotic associations with arthropods, *lptM* and *lptY*, and more rarely *lptE* and *lptD* were lost, suggesting a pathway erosion probably accompanied by a loss of function. Such kind of pathway erosion during the establishment of endosymbiosis has been widely documented³. We have modified Suppl. Fig 2 highlighting with a blue dot the point in the LptY protein tree where *lptY* was acquired, and with a red dot a point where genome reduction led to loss of *lptY*.*

*Concerning whether gene knockouts lead to LPS accumulation in the inner membrane, as indicated in our response 2 to this Reviewer, indeed this is the case. Thus, our results are consistent with the notion that acquisition of *lptM* and *lptY* lead to a gain of function in LPS transport. However, we cannot establish with certainty whether the absence of *lptM* and *lptY* in some genomes implies compensatory mechanisms. We have added a sentence in our discussion to acknowledge this possibility: “Our results show that *LptM* and *LptY* enhance the efficiency of the *Lpt* pathway. However, it remains to be established whether their lack in other bacterial species is offset by compensatory mutations in alternative genes that might contribute to this pathway.”*

4. The bacterial growth experiment in Fig 4e used the streak plate method, while the other related experiments employed the dilution method. It is recommended that the authors standardize the method. On the other hand, the formation of disulfide bonds after mutation was not assessed in Fig 4e, making it impossible to determine whether the protein correctly forms disulfide bonds after the mutation.

*4- We have now performed drop dilution tests that confirm the 4-DPS-dependent growth defect observed for the strains expressing *LptD* variants with the Cys pairs in $\beta 1$ and $\beta 26$ but not in $\beta 1$ and $\beta 2$ (See new table panel in Fig. 4e and the corresponding drop dilution tests illustrated in the new Suppl. Fig. 12a).*

*Addressing the oxidation state of *LptD* with the cysteine pairs in $\beta 1$ and $\beta 26$ is a rather daunting task. In fact, because addition of 4-DPS is lethal in cells expressing these *LptD* variants, it becomes nearly impossible to assess the oxidation state of *LptD*. Nevertheless, taking into account this comment, we have performed several new control experiments. First we have generated another *LptD* variant harbouring a new cysteine pair that can potentially form a disulfide bond between $\beta 1$ and $\beta 26$ as arranged in the contracted state. As the previous two cysteine pair mutants we had already tested, also this new variant strongly inhibits growth only in the presence of 4-DPS (new Supplementary Fig. 12a, *LptD*^{N232C-E757C}). Second, to gain additional evidence that the detrimental effect of 4-DPS is due to the formation of disulfides between $\beta 1$ and $\beta 26$ and is not an indirect consequence of introducing cysteine residues, we have generated 6 additional mutants harbouring only one of the additional cysteines of the pairs, that are N232C, A233C, K234C E757C, I756C, or N755C. Our drop dilution tests show that 4-DPS strongly affects the growth only if *LptD* contains both cysteines of each locking pair and not single cysteines in $\beta 1$ or $\beta 26$ (new Suppl. Fig. 12a). This is an important control that further validates our lateral gate locking strategy. As the single additional cysteine residues have no impact or have only modest impact on growth even in presence of 4-DPS, it appears highly unlikely that the 4-DPS-dependent loss of function with *LptD* N232C/E757C, A233C/I756C or K234C/N755C is caused by interference with translocon biogenesis. Indeed, as addressed in the response point 4 to Reviewer 3, the translocon containing *LptD* N232C/E757C, A233C/I756C or K234C/N755C does not accumulate at the BAM complex (new Suppl. Fig. 12b), thus excluding an indirect detrimental effect due to prolonged occupancy of the BAM complex. Importantly, *LptD* containing the N232C/E757C and K234C/N755C pairs have only a modest impact on the formation of *LptD*^{Ox} whereas lower levels of *LptD*^{Ox} are observed with the A233C/I756C couple (Suppl. Fig 12c of the revised manuscript). Because this variant does not accumulate at the BAM complex, we suspect that A233C/I756C destabilizes *LptD* leading to partial degradation after its biogenesis.*

5. The final sections of the Results contain repetitive descriptions of LptM function. To improve clarity and flow, the authors are advised to streamline these sections and consolidate overlapping findings.

5- Following the suggestion of the Reviewer, we have removed a subheading in the final part of the Results. In particular, we have removed “LptM can be docked from the extended state into the contracted conformation” and incorporated the text under it into the following section entitled “LptM induces gate dynamics, facilitating β 1- β 26 strand separation.”

Reviewer #3 (Remarks to the Author):

The biogenesis of the outer membrane of diderm bacteria requires the transport of bulky lipopolysaccharides (LPS) from the periplasmic face of the inner membrane to the outer leaflet of the outer membrane, a process that is accomplished by the critically important Lpt pathway. Chen and co-workers describe the structural and functional characteristics of LptM and LptY (YedD), two recently discovered accessory factors to the LptD-LptE outer membrane core complex of the LPS translocon. Whilst dispensable under optimal growth conditions, both accessory factors are shown to be required under conditions where Lpt pathway activity becomes limiting. The authors report the cryoEM structure of E. coli LptDE, LptDEY, LptDEMY and LptDEM complexes, and identify two distinct conformational states in LptD, representing an ‘extended’ and ‘contracted’ conformation of the periplasmic b-taco domain and its contact with the lateral gate of the b-barrel domain. These observations are followed up with Cys-Cys and pBpA cross-linking mutants to evaluate the physiological relevance of the conformational states, and their implications for the LPS transport mechanism. Based on these experiments, the authors propose a transport model that invokes conformational cycling between the extended and contracted states to facilitate the passing of LPS molecules from the b-taco to the LptD lumen, and then onwards to the OM outer leaflet.

LPS molecules reach the b-taco domain of the LptDE translocon via the periplasmic LptA bridge and the activity of the inner membrane complex LptB2FG. The mechanism by which the bulky LPS molecules are then translocated to the cell surface and inserted into the OM outer leaflet remains unclear. The discovery and characterization of the LptM and LptY accessory factors will undoubtedly form an important contribution to the functional studies of the LptDE translocon.

This new study is generally well documented and provides solid data to the structural and functional characteristics of LptM and LptY complexes of the LptDE translocon. Although the translocation model proposed by the authors is plausible and in agreement with their experimental data, it remains speculative and would require additional data and/or a more balanced discussion. I’m supportive of publication of this new study provided that some points are clarified by the author.

The main point to address is whether the contracted state of the LptDE taco domain – β 1- β 26 gate is physiologically relevant, or an artefact of a LptDE complex lacking LptM.

1- We understand the concern of this Reviewer, therefore in addition to perform several experiments that further support our proposed mechanisms (see responses below), we have also implemented minor textual revisions in the Abstract, Results and Discussion to adopt a more cautious position. See for instance the modification at the end of the second last paragraph of the Discussion stating: “Additional studies are warranted to comprehensively demonstrate the proposed mechanism of LPS assembly”.

Specific comments:

- By comparison of LptDE and LptDEM and LptDEMY complexes, the authors identify a yet undocumented ‘contracted’ state in the LptDE complex, which in their study correlates with the lack of LptM. From the authors data (e.g. Ext. Data Fig. 9d), it would appear that LptM is a constitutive component of the LptDE translocon. When comparing published LptDE structure from *Klebsiella* and *Shigella*, and the *E. coli* LptDE-YedD complex, the authors show that these fit the extended, LptM-bound state. Did the authors revisit the deposited density maps to check for the presence of unassigned density that may allude to the presence of LptM? Whilst the authors cannot experimentally validate the presence of LptM in these complexes, it would be important to revisit the reported structure and comment on the likelihood they represent LptM bound complexes. i.e. should the field consider LptM a core component of the LptDE complex? It would appear so.

*2- We agree that our data strongly suggest LptM should be considered a core component of the LptDE translocon, in agreement with the following pieces of evidence: (1) constitutive copurification under native conditions (Suppl. Fig. 9d), (2) 1:1:1 stoichiometry in native MS (ref. 36 of the manuscript), (3) assembly during LptD biogenesis at BAM, and (4) functional role in LPS transport shown in our present study. We also examined deposited density maps from *K. pneumoniae* (PDB: 5IV9)⁴, *S. flexneri* (PDB: 4Q35)⁵, and the recent *E. coli* LptDE-YedD structure (PDB: 9FZ5)⁶. While some unassigned density stretches were observed in potential LptM binding regions, we could not reliably attribute these to LptM. The most convincing density was observed in the PDB: 9FZ5 structure (see Suppl. Fig. 8d of the revised manuscript), in which an unassigned density is present in the translocon region where we observe LptM binding. However, we cannot be completely certain this corresponds to LptM, as it could represent other flexible components or lipid molecules. The limited interpretability likely reflects partial translocon occupancy or flexibility. Importantly, LptM has been identified as a subunit of the LPS translocon also in two structures that were published during the revision of our manuscript (PDB: 9KN3)⁷ and (PDB:8HIR)⁸. We have added this information to our manuscript. It should be noted that in the PDB: 9KN3 and PDB: 8HIR LptD is only observed in the extended conformation, highlighting once again the novelty of the contracted state of LptD that we report in our study. In addition, these recent studies do not provide any characterization of the role of LptM during LPS transport, which is instead a key point in our manuscript.*

- Further to previous point, the authors should seek further evidence that the contracted state is physiologically relevant and not an artefact of a LptDE complex lacking LptM. They show that 4-DPS induced oxidation of LptD A233C/I756C and K234C/N755C mutants, which are juxtaposed Cys pairs in the contracted conformation, results in a loss of function. However, can the authors conclude beyond doubt that the loss of function is due to trapping the LptDE(M?) complex in the contracted state versus resulting in a folding defect on the BAM complex?

3- We understand the concern of the Reviewer. Indeed, in the first version of our manuscript we specifically addressed this point by testing the functionality of LptD variants with cysteine couples that would pair specifically in the contracted conformation. Such lateral gate locking strategy was previously employed to test the functionality of the lateral gate of LptD⁹. Our results clearly show that the cysteine pairs that would be juxtaposed in the contracted state lead to a lethal phenotype specifically in presence of the oxidant 4-DPS, even when LptM is produced by the cell. In our opinion this is an important piece of evidence indicating that the contracted state is physiologically relevant and forms also in presence of LptM. Furthermore, we would like to point out that cells rapidly detect and degrade LptD if this is inactive, e.g. due to a reduction of the levels of LPS available for transport¹⁰, or due to a translocon assembly defect as those caused by the expression of the *lptD* allele *imp4213*¹¹ or caused by impaired LptD oxidation due to lack of *DsbA*¹². Hence, it is unlikely that LptD accumulates in a non-functional or non-correctly assembled state as this LptD form would probably be degraded by the cell.

In all cases, following this comment of the Reviewers, we have performed several new control experiments. First we have generated another LptD variant harbouring a new cysteine pair that can potentially form a disulfide bond between $\beta 1$ and $\beta 26$ as arranged in the contracted state. As the previous two cysteine pair mutants we had already tested, also this new variant strongly inhibits growth only in the presence of 4-DPS (new Supplementary Fig. 12a, LptD^{N232C-E757C}). Second, to gain additional evidence that the detrimental effect of 4-DPS is due to the formation of disulfides between $\beta 1$ and $\beta 26$ and is not an indirect consequence of introducing cysteine residues, we have generated 6 additional mutants harbouring only one of the additional cysteines of the pairs, that are N232C, A233C, K234C E757C, I756C, or N755C. Our drop dilution tests show that 4-DPS strongly affects the growth only if LptD contains both cysteines of each locking pair and not single cysteines in $\beta 1$ or $\beta 26$ (new Suppl. Fig. 12a). This is an important control that further validates our lateral gate locking strategy. As the single additional cysteine residues have no impact or have only modest impact on growth even in presence of 4-DPS, it appears highly unlikely that the 4-DPS-dependent loss of function with LptD N232C/E757C, A233C/I756C or K234C/N755C is caused by interference with translocon biogenesis. Indeed, as addressed in the response point 4 to this Reviewer, the translocon containing LptD N232C/E757C, A233C/I756C or K234C/N755C does not accumulate at the BAM complex (new Suppl. Fig. 12b), thus excluding an indirect detrimental effect due to prolonged occupancy of the BAM complex.

- The authors should do a pulldown of the LptD A233C/I756C and K234C/N755C mutants to confirm that these indeed represent fully folded LptDE in contracted state – similar to what is done in Ext. Data Fig. 9d. Moreover, the authors should test for the presence of LptM in case of contraction - locked mutants A233C/I756C and K234C/N755C to address whether the contracted state is compatible with the binding of LptM.

4- In our revised manuscript we have addressed whether the loss of function observed with LptD N232C/E757C, A233C/I756C or K234C/N755C is due to the accumulation of the translocon at the BAM complex. We have performed pull-down of LptDE containing the extra couples of cysteines in LptD $\beta 1$ and $\beta 26$, observing that none of these co-isolate BamD and BamE, differently from what is observed in the absence of LptM (Suppl. Fig. 12b).

In addition, we have tested whether the introduction of the Cys pairs in $\beta 1$ and $\beta 26$ causes dissociation of LptM. To this end, we have compared the amount of LptM that is co-purified

with wild-type LptD or the mutant variants of LptD containing the cysteine couples in $\beta 1$ and $\beta 26$. As shown in Suppl. Fig. 12 b, the introduction of cysteine couples does not impair the association of LptM with LptD. We conclude that the cysteine residues that can potentially lock $\beta 1$ and $\beta 26$ are compatible with LptM binding. Note that these experiments cannot be conducted in presence of 4-DPS as this reagent is lethal for cells that express LptD with Cys pairing in $\beta 1$ and $\beta 26$ as arranged in the contracted state.

- In extended data Fig. 12 the authors show the RMSF of the various LptDE complexes with and without LptM and LptY, both in the extended and contracted states. None of the in silico generated states (LptDE extended conformation, or contracted conformation of LptM containing complexes) relaxes to the state seen experimentally, or shows an increased RMSF in the taco domain and b1-b26 gate. The authors should comment on this point.

5- We have highlighted in the result section that “In all cases, the structures do not switch between extended and contracted states and vice versa, at least over the 500 ns course of the MD simulations”. A conformational switch in the LptD structure may require the establishment of an active transenvelope Lpt bridge or of other physiological conditions that are not taken into account in our MD simulations.

- The authors see an LptM-dependent difference in LPS cross-linking at T236, which they use in support of their argument that LptM increases conformational dynamics of the b-taco – b1-b26 gate contact. How do the authors interpret the lack of LptM responsiveness of the N239 position?

6- A possible explanation of the differential responsiveness of T236 and N239 to the lack of LptM might be related to the different dynamics and positioning of these residues. As highlighted in our original submission, position T236 and N239, present different dihedral angles dynamics (Suppl. Fig. 16 of the revised manuscript). Furthermore, whereas T236 is positioned at the slit of the lateral gate in $\beta 1$, N239 is in $\beta 2$. It appears conceivable that position 236 interacts with LPS most efficiently when LPS passes through the slit of the gate, the dynamics of which is enhanced by LptM. N239, which is positioned more on the inside of the gate, may interact with LPS as soon as this reaches the lumen of LptD β -barrel prior to transport through the gate slit. Thus, the interaction of the latter position with LPS would be unaffected by the reduced gate dynamics in Δ LptM.

- The authors should take a more reserved position in abstract and discussion in the presentation of the proposed mechanistic model. Whilst plausible, an experimental validation of the model is lacking. In their discussion, the authors do not comment on the envisaged function of LptY.

7- In our revised manuscript we have implemented several minor text modifications adopting a more reserved tone when proposing the $\beta 1$ stroke model. Nevertheless, inspired by the comment of the Reviewer we have tested whether amino acid positions in $\beta 26$ also interact with LPS or whether $\beta 1$ is a preferential LPS-interaction site at the lateral gate slit. Interestingly, different from $\beta 1$, positions 751 and 753 in $\beta 26$ did not crosslink LPS (Suppl. Fig; 17), which is in favour of the latter hypothesis. This result provides further support to our model where $\beta 1$ binds LPS to promote its transport. As highlighted in our response 1 to this Reviewer, we clarify in the

discussion that “Additional studies are warranted to comprehensively demonstrate the proposed mechanism of LPS assembly”

Minor points

- Fig. 1. What density is shown, and at what contour level? There is no mention of this in the legend. What is the purpose of the density shown? It does not fit the complexes, or the density shown in Ext Data Fig. 6, 8 or Figure 2, but rather looks like carved out micelles. As is, this is misleading and of little use, or should be explained in Figure / main text.

8- We apologize for the confusion regarding the density representation in Figure 1. The transparent blue surfaces shown represent the DDM micelle densities surrounding the protein complexes, illustrating the membrane-mimetic environment in which the translocon complexes were purified and structurally characterized. We have clarified this in the legends of Figure 1 and Figure 2. We believe that this representation helps readers understand the membrane-embedded nature of these outer membrane protein complexes and provides context for how they would be oriented relative to the bacterial outer membrane.

- Ext. Data Fig. 1a. Presumably the LptDEHis sample results from pLptDE-his expression in a LptM knockout-strain? Clarify this in the legend. The lack of LptM now looks contradictory to Ext. Data Fig. 9d, where expression of pLptDE-his in a WT vs lptM knockout background does result in a strong LptM signal in the IMAC pulldown.

9- The Reviewer is right that the samples LptDEMHis and LptDEHis shown in Fig. 1b and Suppl. Fig. 1a are obtained from transformed wild-type and Δ lptM cells, respectively. In the revised manuscript, we have clarified this important point in the legends of both Fig. 1b and Suppl. Fig. 1a.

- Main text p6 and Ext. Data Fig. 5 describe the co-elution of LptD and LptM with LptY pull-downs. Why is LptE not included in the Western or Coomassie analysis? Presumably the pulled complexes include LptE?

10- We have added this blot, which indeed shows co-purification of LptE together with LptY pull-downs.

- Ext. Data Fig. 5. On what basis do the authors assign the band indicated with * as a non-specific reaction? How come this band indicated as non-specific is lacking in the Ext. Data Fig. 8a?

11- We have tested our antibodies on total cell lysates obtained from strains expressing or lacking LptM. The crossreaction indicated by the asterix is independent of LptM expression. A statement on our antibody testing is included in the Reporting Summary submitted along with our manuscript as required by the Nature Communication editorial policy.

- Ext. Data Fig. 5. It would be helpful to add an anti-His stain for the 15 kDa range.

12- This blot has been added as requested by the Reviewer. The anti-His immunoblot reveals indeed the bait protein LptYHis.

- In p8-9 and Fig. 3 and Ext. Data Fig. 9a, the authors comment on the fitting of the LptY AlphaFold model in the cryoEM map despite their low resolution. Please provide the CC values for the model – map fits.

13- The rigid-body fitting of the LptY AlphaFold2 model into the corresponding densities yielded global correlation values of 0.439 and 0.473 for the LptDEY and LptDEMY cryo-EM maps, respectively, as measured in ChimeraX. These moderate correlation values reflect the lower local resolution (>4.5 Å) in this region of the maps, as shown in Suppl. Fig. 6 (panels f and l). Despite this limitation, we are confident that this low-resolution protrusion corresponds to LptY for several reasons. First, the better-resolved regions near the LptD β-taco and β-barrel domains (local resolution 3–3.5 Å) enabled the ModelAngelo algorithm (Jamali et al., Nature 628, 450–457, 2024) to automatically model YedD loops without prior sequence information, providing unbiased structural evidence. Together with the de novo modelling, LptY was unambiguously identified as a novel interaction partner of LptDE and LptDEM also through native mass spectrometry analysis, which confirmed the presence of LptY as a stoichiometric component of the LPS translocon (as already described in our manuscript, see supplementary Fig. 4).

- Ext. Data Fig. 9d. How do the authors interpret the pulldown of the BamD in absence of LptM, though without the other BAM subunits co-eluting?

14- Both BamD and BamE are co-eluted with LptDE in cells lacking LptM. BamA and BamDE can exist in distinct functional submodules of the BAM complex, that are BamAB and BamCDE, respectively. These submodules can assemble together to form a functional BAM complex¹³. In addition, BamA and BamD bind distinct β-strands of the nascent β-barrel of LptD or of other outer membrane proteins^{14,15}. It is thus conceivable that the stabilized LptDE folding intermediate accumulating in the absence of LptM interacts most stably with BamD rather than with BamA.

- Methods describe the purification of LptDEM and LptDE, but not of LptDEMY or LptDEY. This should be added.

15- Indeed this point needs further clarification. We did not perform a purification specifically for LptDEMY and LptDEY, but these complexes were present in the purification of LptDEM^{His} and LptDE^{His}. To clarify this, we have modified the following sentence in the methods, highlighting that LptDE^{His} contained also LptDE^{His}Y, whereas LptDEM^{His} contained also LptDEM^{His}Y: “The purification of LptDEM^{His} (yielding also LptDEM^{His}Y) or LptDE^{His} (yielding also LptDE^{His}Y) for cryo-EM was performed from 2-liter cultures of a wild-type or a ΔlptM strain carrying respectively pLptDEM^{His} or pLptDE^{His}.”

- Extended Data Fig. 13. Better clarify how the distances after measured. What atoms are used

to calculate the distances and in a residue pair x and y, what is the difference between the x-y and y-x distance?

16- In the mentioned figure (Supplementary Fig. 15 of the revised manuscript), we have calculated distances as follows: x-y is C=O from x and N-H from y; y-x is N-H from x and C=O from y. This means that in the former (x-y) y is the hydrogen bond donor and x the acceptor, whereas in the latter (y-x) x is the hydrogen bond donor and y is the acceptor

We have clarified this point in the corresponding figure legend

Figure: Membrane Deformations of simulated complexes.

Membrane deformations of the phosphorus atoms (vdW spheres) around the simulated protein complexes (grey cartoon at 0 ns) in this study. Phosphate positions are shown for the three repeats of 500 ns and coloured on a blue-white-red scale, where a red colour highlights a membrane thinning, white reflects a standard phosphate-phosphate distance, and blue denotes a thickening of the membrane. Simulations are shown for the five extended (A, C, E, G, and I) and contracted (B, D, F, H, and J) states for (A, B) LptDE, (C, D) LptDEM, (E, F,) LptDEY, (G, H) LptDEMY, and (L, J) LptDEMY with bound KLA. No clear trends in thickening or thinning are observed around the lateral gate region.

References mentioned in our point-by-point response

- 1 Sperandeo, P., Lau, F. K., Carpentieri, A. *et al.* Functional analysis of the protein machinery required for transport of lipopolysaccharide to the outer membrane of Escherichia coli. *Journal of bacteriology* **190**, 4460-4469, doi:10.1128/jb.00270-08 (2008).
- 2 Meredith, T. C., Mamat, U., Kaczynski, Z. *et al.* Modification of lipopolysaccharide with colanic acid (M-antigen) repeats in Escherichia coli. *The Journal of biological chemistry* **282**, 7790-7798, doi:10.1074/jbc.M611034200 (2007).
- 3 McCutcheon, J. P., Garber, A. I., Spencer, N. *et al.* How do bacterial endosymbionts work with so few genes? *PLoS biology* **22**, e3002577, doi:10.1371/journal.pbio.3002577 (2024).
- 4 Botos, I., Majdalani, N., Mayclin, S. J. *et al.* Structural and Functional Characterization of the LPS Transporter LptDE from Gram-Negative Pathogens. *Structure (London, England : 1993)* **24**, 965-976, doi:10.1016/j.str.2016.03.026 (2016).
- 5 Qiao, S., Luo, Q., Zhao, Y. *et al.* Structural basis for lipopolysaccharide insertion in the bacterial outer membrane. *Nature* **511**, 108-111, doi:10.1038/nature13484 (2014).
- 6 Gennaris, A., Nguyen, V. S., Thouvenel, L. *et al.* Optimal functioning of the Lpt bridge depends on a ternary complex between the lipocalin YedD and the LptDE translocon. *Cell reports* **44**, 115446, doi:10.1016/j.celrep.2025.115446 (2025).
- 7 Miyazaki, R., Kimoto, M., Kohga, H. *et al.* Structural basis of lipopolysaccharide translocon assembly mediated by the small lipoprotein LptM. *Cell reports* **44**, 116013, doi:10.1016/j.celrep.2025.116013 (2025).
- 8 Luo, Q., Wang, C., Qiao, S. *et al.* Surface lipoprotein sorting by crosstalk between Lpt and Lol pathways in gram-negative bacteria. *Nature communications* **16**, 4357, doi:10.1038/s41467-025-59660-y (2025).
- 9 Dong, H., Xiang, Q., Gu, Y. *et al.* Structural basis for outer membrane lipopolysaccharide insertion. *Nature* **511**, 52-56, doi:10.1038/nature13464 (2014).
- 10 Martorana, A. M., Moura, E., Sperandeo, P. *et al.* Degradation of Components of the Lpt Transenvelope Machinery Reveals LPS-Dependent Lpt Complex Stability in Escherichia coli. *Frontiers in molecular biosciences* **8**, 758228, doi:10.3389/fmolb.2021.758228 (2021).
- 11 Ruiz, N., Falcone, B., Kahne, D. *et al.* Chemical conditionality: a genetic strategy to probe organelle assembly. *Cell* **121**, 307-317, doi:10.1016/j.cell.2005.02.014 (2005).
- 12 Yang, Y., Chen, H., Corey, R. A. *et al.* LptM promotes oxidative maturation of the lipopolysaccharide translocon by substrate binding mimicry. *Nature communications* **14**, 6368, doi:10.1038/s41467-023-42007-w (2023).

- 13 Hagan, C. L., Kim, S. & Kahne, D. Reconstitution of outer membrane protein assembly from purified components. *Science (New York, N.Y.)* **328**, 890-892, doi:10.1126/science.1188919 (2010).
- 14 Ieva, R., Tian, P., Peterson, J. H. *et al.* Sequential and spatially restricted interactions of assembly factors with an autotransporter beta domain. *Proceedings of the National Academy of Sciences of the United States of America* **108**, E383-391, doi:10.1073/pnas.1103827108 (2011).
- 15 Lee, J., Tomasek, D., Santos, T. M. *et al.* Formation of a β -barrel membrane protein is catalyzed by the interior surface of the assembly machine protein BamA. *eLife* **8**, doi:10.7554/eLife.49787 (2019).

Reviewer #3 (Remarks to the Author):

In response to the reviewers' comments, the authors add an extensive set of new results and additional figure panels. These further strengthen this excellent study and take away most of the uncertainty that was associated with some of the functional models proposed.

A key point to address was the assessment of the physiological relevance of the contracted state of the LptDE translocon, i.e. 1) determining its existence in presence of LptM and 2) determining whether it is an on pathway conformation, or might be an off-pathway artefact from the lptM knockout in which it was identified. The revised manuscript adds experiments to address these points.

Most convincingly, the authors show that the 4-DPS-dependent loss of function of LptD N232C/E757C, A233C/I756C or K234C/N755C, which cross-links b1-b26 in the contracted state, is independent of the presence or absence of LptM, suggesting that the contracted state is also sampled in complexes containing LptM. Furthermore, 4-DPS lethality is dependent on Cys pairs representing the contracted state. Single Cys mutants prove insensitive to 4-DPS. The authors further demonstrate that b1-b26 mutants N232C/E757C, A233C/I756C or K234C/N755C do not accumulate on the BAM complex and are compatible with LptM binding. However, in both cases this was done under reducing conditions, which does not inform on the conformational state of the mutants.

The authors argue these same experiments cannot be done under oxidizing conditions (4-DPS) because of the toxic effect of locking the b1-b26 gate. Can this not be circumvented by adding 4-DPS at higher cell density, so that sufficient cell mass can be harvested for pull-downs and Western analysis? This smaller experiment would still substantially strengthen the authors' proposed model that sampling the LptDE contracted state occurs in fully functional LptDEMY complexes and is part of the translocation activity.

We thank the reviewer for acknowledging that the added results strengthen even further our study. We do not understand, however, the proposal to check the status of LptD in dead cells. The oxidant 4-DPS is toxic in cells expressing LptD with cysteine residues at both sides of its lateral gate, as shown in our drop dilution test on agar plates. We have observed the same growth inhibition effect also when the oxidant is supplemented to liquid cultures. The reviewer's proposal is to increase the cell biomass using a larger number of cells and proceed with the isolation of LptD, despite the fact that we observe no growth. This would mean to collect cells, in which an essential envelope biogenesis machinery has been inactivated leading to an artefactual state of the envelope. We would be unable to withdraw reliable conclusions from such experiment. In any cases, we did attempt to perform purifications from 4-DPS treated cells which yielded extremely low levels of LptE and LptD specifically when we use the mutant strains with cysteines that lock the LptD lateral gate (see figure below). This results testify the impossibility of conducting the experiment proposed by the reviewer.

Importantly, we have acknowledged this limitation of our approach in the revised manuscript as already suggested by this reviewer. In the Discussion, we clearly state: "Additional studies are warranted to comprehensively demonstrate the proposed mechanism of LPS assembly".

Figure Legend

*Cells deleted of chromosomal *lptD* and harboring plasmid-borne wild-type *lptD* or mutated *lptD* alleles (A233C/I756C or K234C/N755C) were cultured and supplemented with 0.05 mM 4-DPS for 1h before cell harvesting. Collected cells were subjected to envelope fractionation, solubilization with DDM followed by nickel-affinity purification of LptE^{His}. Elution fractions were subjected to SDS-PAGE and coomassie blue staining.*